# Chemoproteomic capture of RNA binding activity in living cells

Andrew J. Heindel[1], Jeffrey W. Brulet[2], Xiantao Wang[3], Michael W. Founds[2], Adam H. Libby[2,4], Dina L. Bai[2], Michael C. Lemke[1], David M. Leace [1], Thurl E. Harris [1], Markus Hafner [3] & Ku-Lung Hsu [1,2,4,5,6]

Proteomic methods for RNA interactome capture (RIC) rely principally on crosslinking native or labeled cellular RNA to enrich and investigate RNA-binding protein (RBP) composition and function in cells. The ability to measure RBP activity at individual binding sites by RIC, however, has been more challenging due to the heterogenous nature of peptide adducts derived from the RNA-protein crosslinked site. Here, we present an orthogonal strategy that utilizes clickable electrophilic purines to directly quantify protein-RNA interactions on proteins through photoaffinity competition with 4-thiouridine (4SU)-labeled RNA in cells. Our photo-activatable-competition and chemoproteomic enrichment (PACCE) method facilitated detection of >5500 cysteine sites across ~3000 proteins displaying RNA-sensitive alterations in probe binding. Importantly, PACCE enabled functional profiling of canonical RNA-binding domains as well as discovery of moonlighting RNA binding activity in the human proteome. Collectively, we present a chemoproteomic platform for global quantification of protein-RNA binding activity in living cells.

RNA-binding proteins (RBPs) constitute a large (~7% of the human proteome) and diverse class of proteins that control RNA metabolism and function[1–3]. RBPs bind sequence and/or structural motifs in single- or double-stranded RNA using RNA-binding domains (RBDs) as constituents of ribonucleoprotein (RNP) complexes that regulate gene expression and non-coding RNA (ncRNA) function[4,5]. Canonical RBDs include, for example, RNA recognition motif (RRM), DEAD box helicase, and hnRNP K homology (KH) domains, which mediate binding affinity, avidity and specificity of target RNA sequences[6–10]. RNPs regulate gene expression and ncRNA function through dynamic RNA-protein networks to control biogenesis, transport, translation, and degradation of bound RNAs[5,11]. Although large compendiums of RBPs have been inventoried by proteomics (i.e., the RNA interactome[12,13]), corresponding methods for direct quantification of RNA binding sites on proteins for activity and inhibitor profiling are lacking.

Widely used proteomic methods for RNA interactome capture (RIC) deploy ultraviolet light (UV) irradiation to cross-link RBPs to polyadenylated (poly(A)) RNA in cells followed by oligo(dT)-mediated purification of RNPs and tandem liquid chromatography-mass spectrometry (LC-MS/MS) identification. RBP crosslinking and immunoprecipitation in cells can be achieved through UV-excitation of native nucleoside bases (254 nm[14]) or using a photoactivatable ribonucleoside analog 4-thiouridine (4SU[15]) that is metabolically incorporated into labeled RNAs (365 nm) for RIC[12,13,16]. Variations on this technique[17–19] include metabolic labeling of RNAs with an alkynyl uridine analog combined with 4SU crosslinking for RIC of RBPs on non-

[1]Department of Pharmacology, University of Virginia School of Medicine, Charlottesville, VA 22908, USA. [2]Department of Chemistry, University of Virginia, Charlottesville, VA 22904, USA. [3]RNA Molecular Biology Laboratory, National Institute of Arthritis and Musculoskeletal and Skin Disease, Bethesda, MD 20892, USA. [4]University of Virginia Cancer Center, University of Virginia, Charlottesville, VA 22903, USA. [5]Department of Molecular Physiology and Biological Physics, University of Virginia, Charlottesville, VA 22908, USA. [6]Present address: Department of Chemistry, University of Texas at Austin, Austin, TX 78712, USA. ✉e-mail: ken.hsu@austin.utexas.edu

poly(A) RNAs (CARIC[20]). Methods based on differential solubility of RNPs have also been pursued for RBP investigations[21,22]. Collectively, these methodologies have identified hundreds of RBPs as regulatory components of RNPs mediating cell differentiation, embryonic development, inflammation, and viral sensing[1-3].

An intriguing finding from RIC experiments is the observation that a large fraction of identified proteins in yeast and human cells lack canonical RBDs or were not previously assigned a role in RNA biology[1]. These unannotated RBPs (designated as 'enigmRBPs') are enriched for metabolic enzymes with glycolysis as a particular hotspot for enzymes with RNA-binding function[23]. GAPDH, for example, was previously validated as an authentic RBP in post-transcriptional regulation of T cell effector function[24]. An important step towards discovering non-canonical RBPs are methodologies for unbiased identification of RNA-binding regions on proteins by LC-MS/MS. Direct identification of RNA crosslinked sites on proteins has been demonstrated; however, this method (i.e. RNP[xl]) is typically lower in sensitivity and requires specialized computational workflows to address the heterogenous character of peptide-RNA oligonucleotide conjugates detected[25]. Lower resolution methods for identifying RNA-binding regions (~17 amino acids) include a variant of RIC that incorporates a protease digestion step prior to a second round of oligo(dT) capture to identify peptides flanking the crosslinked site for in silico reconstruction of the RNA-bound regions on RBPs (RBDmap[26]). The mass shift resulting from RNA crosslinked to peptides (~9 amino acids) has been used to infer the location of RNA-bound sites through depletion of tryptic peptide LC-MS/MS signals in UV- versus non-irradiated controls (RBR-ID[27]). Despite the success of the aforementioned methods, current techniques for investigating RBDs sacrifice either sensitivity or binding site resolution[1].

Here, we develop a Photo-Activatable-Competition and Chemo-proteomic Enrichment (PACCE) method for global quantification of protein-RNA interactions in living cells. PACCE is differentiated from available RIC methods by deploying chemical probes to covalently bind, enrich, and identify the human RNA-binding proteome. Importantly, PACCE can detect protein-RNA interfaces with amino acid resolution by quantifying sensitivity of probe-modified sites to competition with photoactivatable cellular RNA. Using PACCE, we assessed >5500 RNA-sensitive cysteine sites that mapped to a large fraction of proteins mediating recognition of coding and noncoding RNA. PACCE enabled functional profiling of RNA-binding regions on known RBPs as well as discovery of moonlighting RNA binding activity in situ.

## Results

### Development of clickable electrophilic purines

We reasoned that the purine heterocycle is well suited for developing a chemoproteomic probe because the fused pyrimidine-imidazole aromatic ring system affords both electron-deficient and -rich sites for integration of electrophilic and reporter tags (Fig. 1A). The π-deficient pyrimidine ring of purine contains electron-deficient carbons at the C2 and C6 positions that can be converted into an electrophilic site for nucleophilic aromatic substitution ($S_NAr$) reactions with protein nucleophiles[28] (Supplementary Fig. 1A). Activation of the C6 position for $S_NAr$ reaction using various leaving groups[29-32] including halogens[33,34] has been demonstrated in synthetic chemistry. We selected a chloro-leaving group because of its tempered ability to activate purines for $S_NAr$ reaction[35]. An alkyne reporter tag was appended at the N7 or N9 position, and the regioisomeric products were confirmed by X-ray crystallography to generate the clickable electrophilic purines (CEPs) AHL-Pu-1 and AHL-Pu-2, respectively (Supplementary Fig. 1B).

We confirmed by HPLC[36] that CEPs undergo $S_NAr$ reaction with nucleophiles that mimic amino acid sidechain groups. Both probes reacted with the cysteine mimetic butanethiol in a time-dependent manner with AHL-Pu-1 showing a modest enhancement in reactivity compared with AHL-Pu-2 ($t_{1/2} = 1.9$ and 9.1 min, respectively; Supplementary Fig. 1B, C, 2). Neither probe showed reactivity against the other nucleophiles tested except for moderate activity of AHL-Pu-1 for

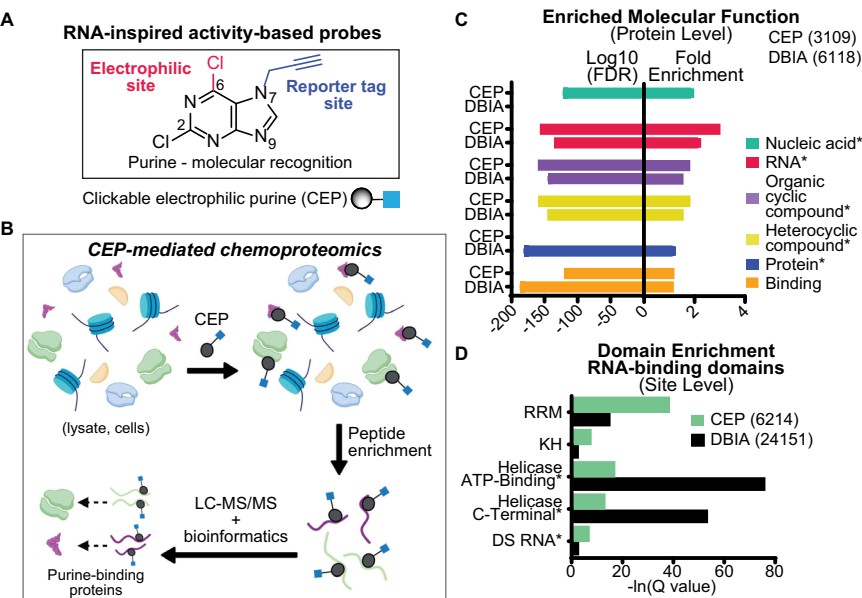

**Fig. 1 | Development of clickable electrophilic purines for chemical proteomic profiling. A** The purine base is a core component of RNA and other biologically important molecules and inspired the design of covalent probes for activity-based profiling. The electronic features of the purine heterocycle were used to install electrophilic and reporter tag groups for developing clickable electrophilic purines (CEPs). **B** CEPs facilitate the enrichment and identification of purine-binding proteins from complex samples (lysates, cells) using standard chemoproteomic workflows for protein and binding site identifications. Graphic created using BioRender (https://www.biorender.com). **C** Gene Ontology (GO) analyses identified RNA binding as an enriched function from chemoproteomic analyses of CEP (AHL-Pu-1)-treated cells. GO analysis of DBIA datasets were included as a comparison. The top 5 domains based on FDR are displayed. GO comparisons were performed as described in Supplementary Methods. **D** Aggregate datasets for AHL-Pu-1 and DBIA binding activity are enriched for RNA-binding domains (RBDs) detected in cell proteomes. Asterisk denotes binding function.

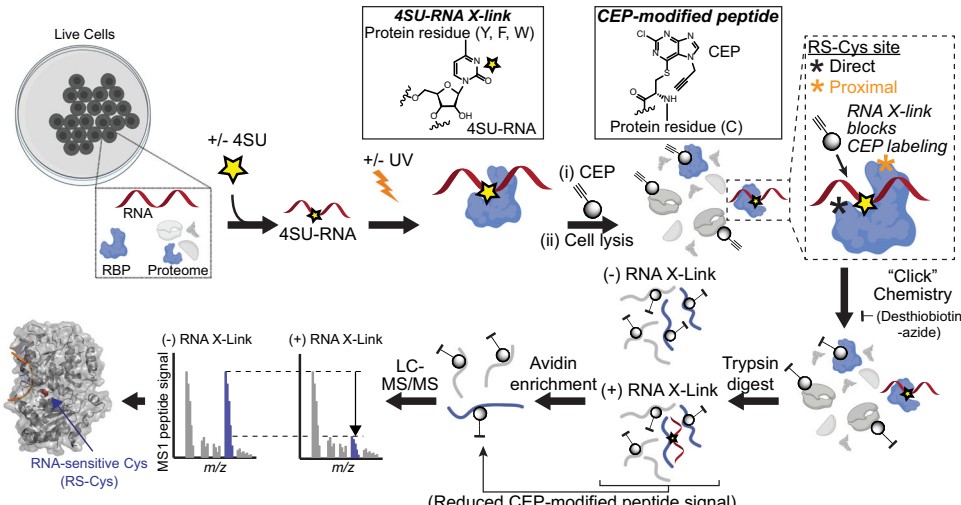

**Fig. 2 | Capturing RNA-binding activity of proteins in situ using photoaffinity competition.** Schematic of Photo-Activatable-Competition and Chemoproteomic Enrichment (PACCE) methodology. UV crosslinking stabilizes cellular RBP-RNA interactions and blocks CEP probe binding to RNA-sensitive cysteine sites (RS-Cys shown in inset) found in protein-RNA interfaces proteome-wide. Homogeneous probe adducts produced from covalent binding of CEP to cysteine sites on proteins ($S_NAr$) compared with UV-mediated crosslinking of heterogenous cellular 4SU-RNA to various residues on protein sites are shown in insets. See Supplementary Fig. 10 and Supplementary Methods for additional details on PACCE conditions and workflow. Graphic created in BioRender (https://www.biorender.com).

---

*p*-cresol (Supplementary Fig. 2). The structure of reaction product peaks observed by HPLC were confirmed by comparison with synthetic standards that were analyzed by X-ray crystallography and NMR (Supplementary Fig. 1D). We also tested stability of CEPs in an aqueous buffer and observed negligible degradation after incubation for 2 days (Supplementary Fig. 2C). See Supplementary Information for additional details of CEP synthesis and characterization.

### Benchmarking CEPs as cysteine-reactive probes in living cells

Concentration- and time-dependent evaluation of CEPs was performed in live DM93 cells to identify non-toxic treatment conditions with minimal perturbation to the transcriptome and proteome (Supplementary Figs. 3 and 4). Next, SILAC light and heavy DM93 cells were treated with dimethyl sulfoxide (DMSO) vehicle or CEP (25 μM, 4 h) followed by cell lysis and copper-catalyzed azide-alkyne cycloaddition (CuAAC) conjugation of desthiobiotin-azide to CEP-modified proteins, avidin chromatography enrichment of CEP-modified peptides, and high-resolution LC-MS/MS and bioinformatics analysis as previously described[37] and depicted in Supplementary Fig. 5. Consistent with our HPLC findings, both CEPs exhibited high chemoselectivity for cysteines; ~72% of probe-modified residues detected in CEP datasets were assigned to cysteines (mass adduct of 604.2637 Da, Supplementary Fig. 6). We verified that CEP binding activity was purine dependent by demonstrating that CEP-modified sites were largely competed with free purine but not pyrimidine nucleobases in competition studies in vitro (Supplementary Figs. 5 and 7). The position of the alkyne functional group at the N7 (AHL-Pu-1) versus N9 (AHL-Pu-2) position affected CEP binding activity; we selected AHL-Pu-1 for the remaining studies because of broader proteome reactivity with similar protein class coverage compared with AHL-Pu-2 (Supplementary Fig. 8, Supplementary Data 1).

CEP-mediated chemoproteomics was applied to additional human adherent and suspension cell lines to quantify thousands of probe-modified cysteine sites on proteins in situ. Gene Ontology[38,39] (GO) analyses of probe-modified proteins from aggregate AHL-Pu-1-treated cell datasets revealed enrichment for proteins involved in nucleic acid-, RNA- and general heterocyclic compound-binding (Fig. 1B–C, Supplementary Data 2 and 3). Compared with the general cysteine-reactive probe iodoacetamide (IA) and specifically datasets using the desthiobiotin-tagged analog (DBIA[40]), protein function

enrichments were largely comparable between probes with the exception of nucleic acid- and protein-binding that were specific for CEP and DBIA, respectively (Fig. 1C).

The enrichment for RNA binding function prompted further examination of CEP coverage of this protein class given the growing interest in developing small molecule binders of RBPs[41]. We observed statistically significant overlap with reported human RBPs[1,12,13,20] using CEP-mediated chemoproteomics (~37% overlap, $p = 4.09 \times 10^{-5}$; Supplementary Fig. 8D, Supplementary Data 2 and 3). Both CEP and DBIA showed substantial coverage of detected RBPs (~37 and 61%, respectively) with an additional ~130 RBPs captured using the former cysteine-reactive probe (Supplementary Fig. 8D). Domain enrichment[36] analyses also revealed comparable coverage of RBDs, as well as other functional protein domains, for both CEP and DBIA (Fig. 1D and Supplementary Data 3).

In summary, we demonstrate CEPs are non-toxic, cysteine-reactive probes that are complementary to existing IA probes but can serve as effective chemoproteomic capture agents for global protein- and binding-site level quantification of RBPs in live cells.

### Quantifying protein-RNA interactions in cells by photoaffinity competition

We established a Photo-Activatable-Competition and Chemoproteomic Enrichment (PACCE) strategy to identify CEP-modified sites on proteins that are competed by RNA photo-crosslinking competition (RNA-sensitive cysteine or RS-Cys site, Fig. 2). The photoactivatable nucleoside 4SU is metabolically incorporated into labeled RNAs in cell culture to facilitate UV crosslinking of photoreactive 4SU-RNA to RBPs in situ[13,15]. We chose to incorporate 4SU for UV-mediated crosslinking of cellular RNA to protein at 312 nm because of (i) higher specificity (e.g., DNA-protein crosslinks and single-strand breaks are ~1000-fold less at 312 compared with 254 nm[42]) without compromising proteomic sensitivity[27,43], (ii) less damage to cells[1,27], and (iii) a UV wavelength closer to the optimum extinction coefficient of 4SU[44] (Supplementary Figs. 4 and 9).

PACCE can globally capture RS-Cys located in or proximal to the RNA crosslinked site by identifying direct and proximal competition events (SILAC ratio or SR > 2) from both 4SU-dependent and native RNA crosslinking (Supplementary Fig. 10A). First, we tested whether CEP labeling in vitro vs in situ impacted the ability to

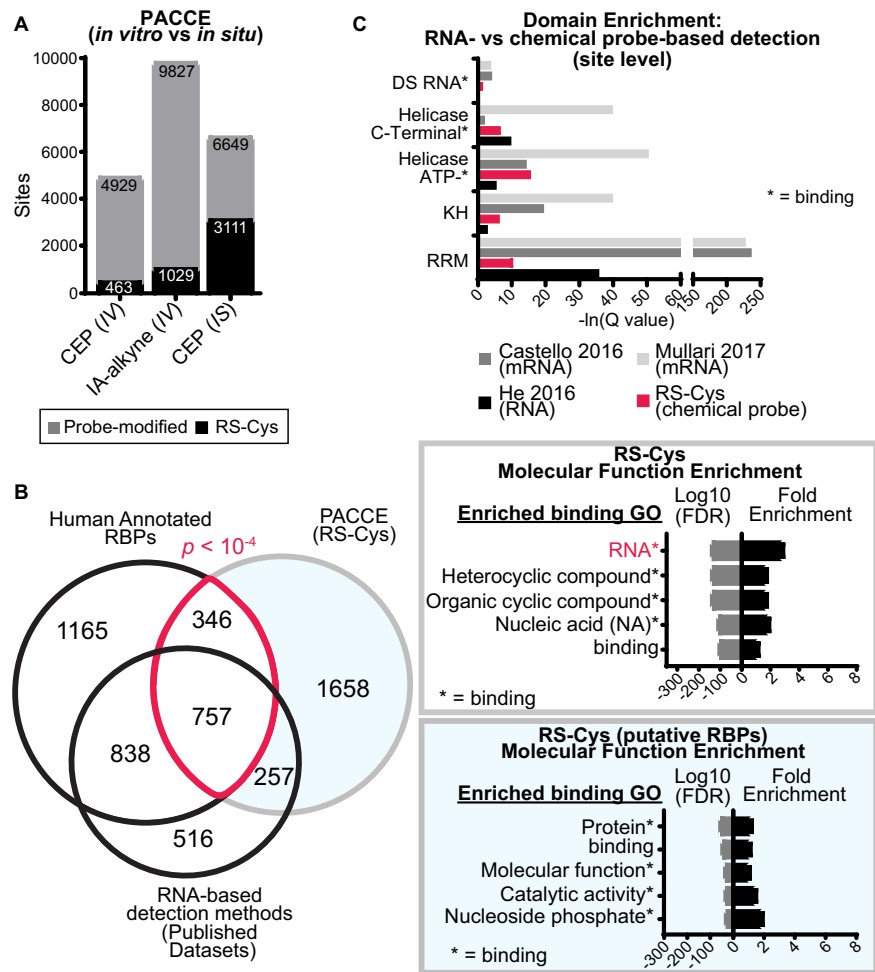

**Fig. 3 | Quantifying RNA-sensitive cysteine sites in situ using PACCE.**
**A** Comparison of in vitro (IV) vs in situ (IS) CEP probe labeling for PACCE. IA-alkyne is included as a general, cysteine-reactive probe counterpart for benchmarking PACCE. Data shown are derived from HEK293T cells. **B** Proteins containing RS-Cys sites (aggregate from PACCE in situ analysis of DM93 and HEK293T cells) were compared with proteins detected by RNA-based proteomic capture methods and the human RBP database[20]. Significant overlap between RS-Cys-containing proteins and human RBPs was determined by a hypergeometric test as described in the Supplementary Methods ($p = 2.97 \times 10^{-5}$). RS-Cys-containing proteins (top inset) were enriched for RNA and nucleic acid binding as determined by GO. RS-Cys-containing proteins that did not overlap with annotated human RBPs (putative RBPs, bottom inset) were enriched for general binding functions. **C** Domain enrichment analysis of aggregate RS-Cys sites (DM93 and HEK293T) compared with the inferred binding regions from RNA-based proteomic detection methods revealed prevalent binding at RBDs. Enriched domain assignments are those with a $Q < 0.01$ after Benjamini–Hochberg correction of a two-sided binomial test.

detect RS-Cys sites because RBPs are known to function in larger RNP complexes that can be disrupted upon cell lysis[1]. SILAC HEK293T cells were subjected to RNA crosslinking competition followed by in vitro CEP labeling (25 μM, 1 h). An alkyne-tagged iodoacetamide probe counterpart was included for direct comparison (IA-alkyne; 500 μM, 1 h). As expected, IA-alkyne captured ~2-fold the total number of cysteines compared with CEP, which resulted in a comparable increase in the number of detectable RS-Cys sites (Fig. 3A). In contrast and in support of using cell-active probes for maximal RS-Cys detection, CEP activity in situ (25 μM, 1 h) produced only a modest increase in total cysteine site coverage (~35%) but a nearly 7-fold enhancement in RS-Cys sites captured (Fig. 3A and Supplementary Fig. 11). RNA crosslinking competition was further supported by a substantial reduction in the number of RS-Cys sites detected when RNase was added prior to UV irradiation of proteomes (Supplementary Fig. 12).

Using RNA crosslinking competition and in situ CEP labeling, a compendium of RS-Cys sites was produced from analyses in HEK293T and DM93 cells, which were cell lines selected based on a high number of CEP-modified sites identified. In aggregate, we detected 11,385 CEP probe-modified peptides corresponding to 4,523 protein identifications. From this dataset, we identified >5000 RS-Cys sites that mapped to ~3000 proteins (Supplementary Fig. 10A–C and Supplementary Data 4). The mean SR for RS-Cys peptides were generally higher compared with non-RS-Cys counterparts (Supplementary Fig. 10D). Approximately 37% of RS-Cys-containing proteins are known RBPs and this number of RS-Cys-containing RBPs represent ~36% coverage of the human annotated RBPs[1,3] (~36% overlap, $p = 2.97 \times 10^{-5}$; Fig. 3B and Supplementary Fig. 13).

The RS-Cys-containing proteins were enriched for functional terms related to RNA binding and metabolism (Fig. 3B and Supplementary Data 5). Importantly, analysis of RS-Cys sites revealed prominent domain enrichment of known RBDs that was comparable with published RNA-based proteomic detection methods for RBP analyses[26,27,45] (Fig. 3C). We also identified non-canonical RNA binding regions within known RBPs (Supplementary Data 5). The frequency of these putative non-canonical RNA-binding regions varied between RBP class and even within members of the same RBP superfamily as exemplified by the DEAD box RNA helicase family[9] (Supplementary Data 6). Analysis of our PACCE datasets also revealed substantial modification of intrinsically-disordered regions that combined with our domain enrichment analyses

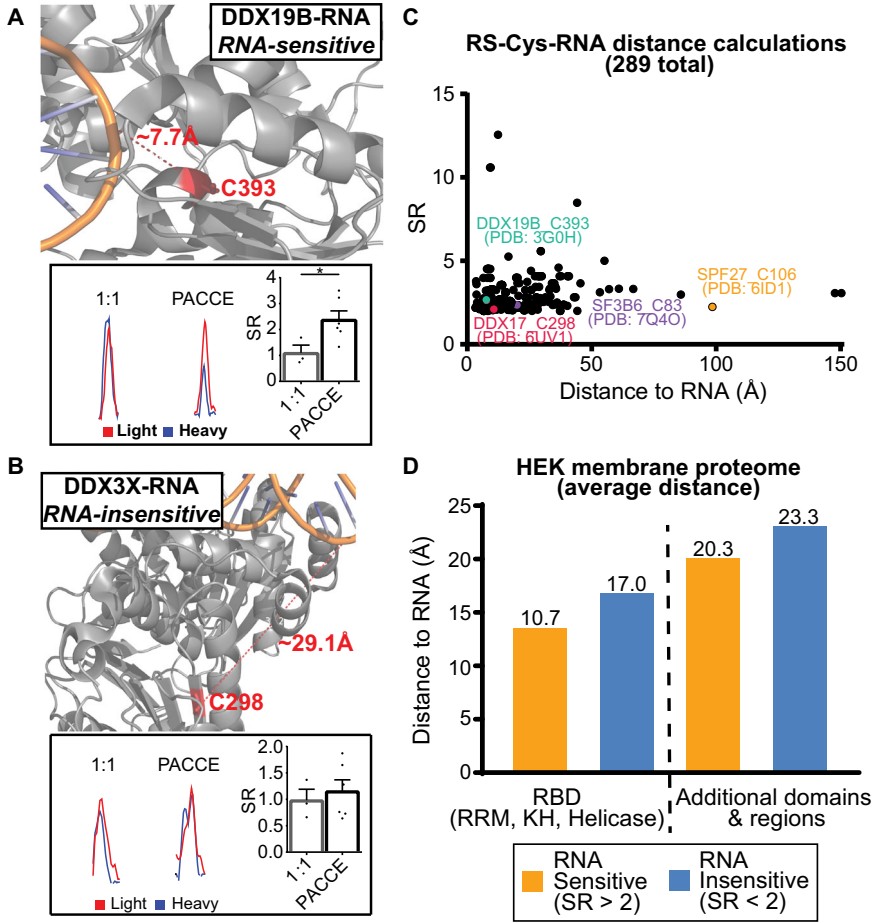

**Fig. 4 | Location of RNA-sensitive cysteine sites in protein-RNA interfaces.**
**A** Location of a RS-Cys detected on DDX19B (C393, PDB ID: 3G0H) in proximity to bound RNA (8 angstroms). Representative MS1 EICs demonstrating significant blockade of CEP labeling at DDX19B C393 (SR > 2) from RNA crosslinking (312 nm) vs. the 1:1 mixing control for equivalent CEP labeling and mixing of SILAC light (red) and heavy proteomes (blue) using a one-tailed Mann–Whitney U-test ($p = 0.0238$). DDX19B probe-modified peptide is shared with DDX19A. **B** DDX3X C298 is not sensitive to RNA crosslinking (red XICs = SILAC light, blue XICs = SILAC heavy), which agrees with its larger calculated distance (29 angstroms) to the bound RNA (PDB ID: 6O5F). Statistics were calculated using a one-tailed Mann–Whitney U-test

($p = 0.4524$). **C** Plot of aggregate RS-Cys SR values (DM93 and HEK293T) as a function of the distance between the respective site and interacting RNA across RBP-RNA structures analyzed (289 structures in total). **D** The distance between the quantified site and bound RNA of RBP-RNA structures is generally reduced for RS-Cys (SR > 2). Data shown are sites from HEK293T membrane proteomes subjected to RNA crosslinking competition (4SU- and native-RNA). RNA insensitive sites include non-probe modified Cys-sites. Data shown are mean ± SEM for $n = 3$ biologically independent replicates for 1:1 mixing sample and $n = 6$ independent replicates for PACCE. *$p < 0.05$. Details on analyses can be found in Supplementary Methods.

identified ~900 protein domains or regions with annotated RNA binding function that contain a RS-Cys site (Supplementary Data 5 and 7).

In summary, PACCE is capable of quantifying protein-RNA interactions directly in live cells by integrating established photoactivatable ribonucleosides with chemoproteomic workflows (Supplementary Fig. 14). The discovery of RS-Cys sites in known and non-canonical RNA-binding regions provides additional opportunities for covalent binding to cysteines for pharmacological modulation of RBPs as demonstrated in a recent report[46].

**Location of RS-Cys sites in protein-RNA interfaces**
Closer inspection of RS-Cys site and proximity to bound RNA provided additional clues to the observed sensitivity of these sites to RNA crosslinking competition. For example, the RS-Cys detected on DDX19B (C393) and DDX17 (C298) are located close to the bound RNA and this proximity is reflected in the sensitivity of the respective sites to RNA crosslinking (SR > 2 in PACCE compared to ~1 for the mixing control, $p \leq 0.05$; Fig. 4A, Supplementary Figs. 15 and 16). As a direct comparison, the RNA-insensitive site detected on DDX3X was not in proximity to bound RNA, which agrees with the observed lack of

competition from RNA crosslinking (C298, SR ~ 1 for the mixing control and RS-Cys conditions; Fig. 4B).

We searched 270 protein-RNA structures (X-ray and cryo-electron microscopy) available in the Protein Data Bank (PDB)[47] to broaden our evaluation of RS-Cys location in proximity to bound RNA on RBPs. The distance from the thiol group of all cysteine sites to the nearest atom on the RNA molecule in structures were calculated using an in-house algorithm. From this group of cysteines, we identified 60 RBP-RNA structures that contained a RS-Cys site. We used these data to assess RNA sensitivity by PACCE (SR value) as a function of distance between the CEP-modified cysteine and interacting RNA (Fig. 4C). The co-crystal structures, CEP-modified sites, PACCE status, and calculated Euclidean distances are available in Supplementary Data 7.

The mean cysteine-RNA distance across all RBP-RNA structures analyzed was ~23 angstroms and this distance was reduced to ~20 angstroms in RS-Cys sites. If we consider RS-Cys sites found in known RBDs, this mean cysteine-RNA distance is further lowered (~11 angstroms, Fig. 4D). We also found examples of RS-Cys located at a larger than expected distance from the RNA interaction site. For example, the C83 site in the RRM domain of SF3B6 had a calculated cysteine-RNA distance of 20 angstroms (Fig. 4C and Supplementary Fig. 15). This

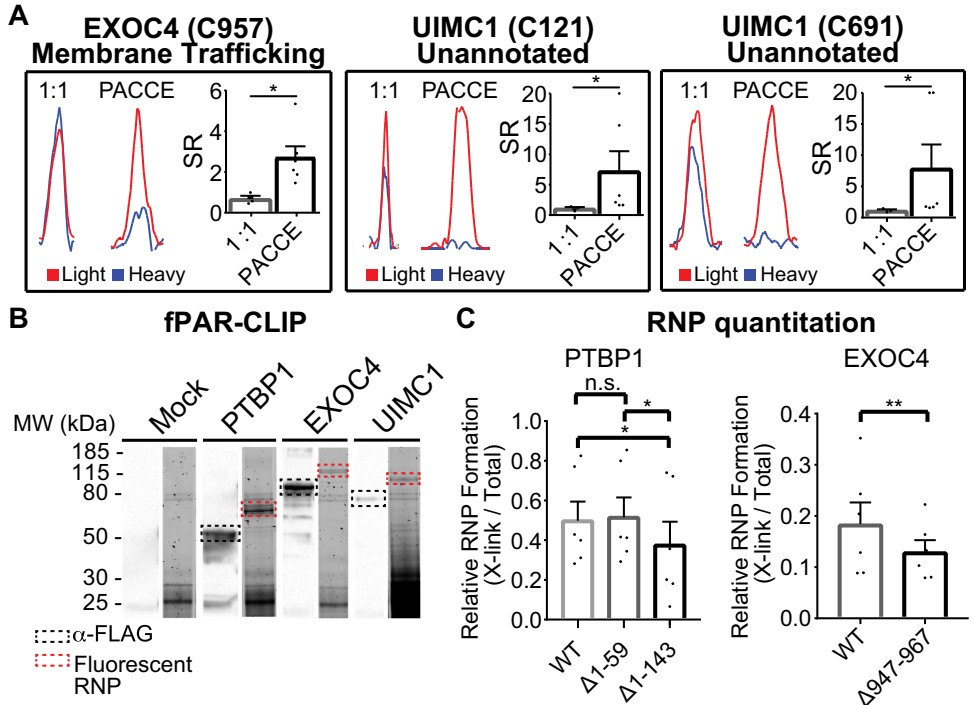

**Fig. 5 | Discovery of moonlighting RNA-binding activity in proteomes.**
**A** Representative MS1 EICs for EXOC4 (C957: mean SR = 2.7 vs. 0.7, $p$ = 0.0119) and UIMC1 (C121: mean SR = 7.2 vs 1.1, $p$ = 0.0119; C691: mean SR = 7.9 vs 1.1, $p$ = 0.0119 for PACCE compared with the SILAC light/heavy 1:1 mixing control, respectively) RS-Cys sites. Statistics were calculated using a one-tailed Mann–Whitney $U$-test. Red XICs represent SILAC light, while blue XICs represent SILAC heavy proteomes.
**B** Fluorescent image of SDS-PAGE separating fluorescent-adapter ligated, cross-linked FLAG-PTBP1, -EXOC4, and -UIMC1 RNPs. Bands boxed in red correspond to the molecular weight of the FLAG-tagged protein plus the fluorescently labeled adapter. Counterpart western blots (α-FLAG, black box) confirmed expression of recombinant protein. **C** Western blot analyses comparing RNP formation of wild-type (WT) and corresponding RBP mutants. Full length blots can be found in Supplementary Fig. 20C. Integrated band intensities from the WT and mutant RNP bands were used to quantify the impact of deleting RNA-binding domains or regions on RNP formation. Data shown are mean ± SEM; $n$ = 3 biological replicates for 1:1 mixing condition LC-MS studies, $n$ = 6 biologically independent replicates for PACCE LC-MS studies, $n$ = 6 biologically independent replicates for western blots. *$p$ < 0.05; **$p$ < 0.01. EXOC4 was analyzed using a ratio paired $t$-test (paired, parametric, one-tailed, $p$ = 0.0035), while PTBP1 was analyzed using a one-way ANOVA ($p$ = 0.0118) followed by a Tukey's post hoc test for multiple comparisons ($p$ = 0.9007, 0.0309, 0.0151, respectively).

longer distance suggests dynamic RBP-RNA interactions[48–50] that are difficult to capture in static structures but can be revealed by PACCE (Supplementary Fig. 15). A subset of RS-Cys sites with longer distances from RNA were detected on proteins found in large multi-RBP complexes that made it difficult to evaluate individual RBP-RNA distances (e.g., SPF27 C106, 98 angstroms; Fig. 4C and Supplementary Data 7).

**Discovery of moonlighting RNA-binding activity in the human proteome**

A substantial fraction of the RS-Cys sites mapped to proteins not found in the annotated RBP interactome ('putative RBPs' group, Fig. 3B). Analysis of this group of proteins revealed enrichment for terms related to molecular- (e.g., ligand binding) and protein-binding. The ligand binding group included E3 ligases (PPIL2), ubiquitin-binding proteins (UIMC1), and structural proteins (EXOC4). Interestingly and in further support of PACCE for RBP profiling, PPIL2 was recently demonstrated to exhibit RNA-binding activity as a component of the minor spliceosome[51]. To test whether PACCE could discover 'moonlighting' RBP activity[23], we selected candidate proteins that contained at least a single RS-Cys and had not been previously annotated as a RBP.

We identified the exocyst complex component 4 (EXOC4) protein as a candidate for our proof-of-concept studies because it contained a single RS-Cys located in an unknown region (C957, Fig. 5A and Supplementary Fig. 17A). EXOC4 is a one of the eight subunits of the exocyst protein complex, which functions to tether post-Golgi secretory vesicles to the plasma membrane before exocytic

fusion[52,53]. Previous studies reported protein-protein interaction function for EXOC4[54] but only a very limited number of reports described nucleic acid binding[55,56]. Akin to the migration behavior of a known RBP (Polypyrimidine tract-binding protein 1 or PTBP1), we observed a shift in molecular weight upon UV irradiation of 4SU-labeled, EXOC4-expressing HEK293T cells that was muted in the absence of UV irradiation and reduced with RNase treatment (Supplementary Fig. 18A, B). The higher molecular weight signals observed without UV irradiation are likely due to crosslinking from ambient light as previously reported[57].

We chose a functionally orthogonal candidate, BRCA1-A complex subunit RAP80 (UIMC1) to demonstrate that PACCE can discover unanticipated RBP activity across different protein classes. UIMC1 is annotated as a ubiquitin-binding protein that recognizes ubiquitinated histones found at sites of DNA damage to direct the BRCA1-BARD1 complex to repair DNA double-strand breaks (DSBs)[58]. We identified RS-Cys sites in the ubiquitin-interacting motif (UIM, C121) and unknown region of UIMC1 (C691, Fig. 5A and Supplementary Fig. 19). The UIM domains facilitate UIMC1 recognition of Lys[63]-linked poly-ubiquitin chains at DNA damage sites[59–61]. The evolutionary conservation of these cysteines further support function (Supplementary Fig. 19A). Photoactivated crosslinking of 4SU-RNA in UIMC1-expressing HEK293T cells resulted in RNase-sensitive, gel migration behavior that supports RBP activity for UIMC1 (Supplementary Fig. 18C).

Next, we tested whether our candidate RBPs, EXOC4 and UIMC1 directly bound RNA in cultured cells by fluorescent photoactivatable ribonucleoside-enhanced crosslinking and immunoprecipitation

(fPAR-CLIP[62]). We transfected HEK293T cells with plasmids encoding FLAG-tagged EXOC4, UIMC1, and PTBP1 (RBP control), labeled nascent RNA with 4-SU and crosslinked RNPs with 312 nm UV. Next, we immunoprecipitated the FLAG-tagged RNPs, and ligated fluorescent oligoribonucleotide adapters to interacting RNAs followed by SDS-PAGE (Figs. 5B, S18D and E). Fluorescently labeled FLAG-RNPs for all three proteins migrated at ~25 kDa above their predicted size, corresponding to the molecular weight of the fluorescent adapter, indicating that not only the canonical RBP, PTBP1 bound RNA in cells, but also our candidates UIMC1 and EXOC4.

We further verified protein-RNA interactions by the polynucleotide kinase (PNK) assay[63]. Cellular lysates from UV (312 nm) irradiated, 4SU-RNA-treated HEK293T cells recombinantly expressing PTBP1, EXOC4 or UIMC1 were subjected to RNaseA at increasing concentrations. Afterwards, proteins were immunoprecipitated followed by radioactive labeling ($^{32}$P) of RNA 5′ ends with T4 PNK. As shown in Supplementary Fig. 18F, we observed increased higher molecular weight radiolabeled bands upon UV irradiation of HEK293T cells overexpressing RBP and this expected 'smeared' signal corresponding to RNP complexes was reduced in a RNaseA-concentration dependent manner.

Collectively, we demonstrated that PACCE can discover RNA binding activity for proteins without prior RBP annotation. The unbiased nature of PACCE is further highlighted by the identification of RS-Cys sites in poorly defined regions (EXOC4) or in domains lacking RBD annotation (UIM domain, UIMC1).

### Validation of EXOC4 C957 role in RNP formation

Next, we tested whether deleting protein regions containing a RS-Cys site affects RNP complex formation in 4SU-RNA crosslinked cells. We used recombinant PTBP1 for proof of concept that deletion of RNA-binding regions on a known RBP resulted in quantifiable changes in crosslinked RNP species. PTBP1 contains 4 RRM domains that are reported to function in RNA binding[64,65]. We recombinantly expressed PTBP1 mutants that progressively deleted the N-terminal region containing RRM1 in HEK293T cells (PTBP1Δ1-59 and Δ1-143 mutants) and assessed the resulting effects on PTBP1 RNP complexes formed in cells compared to WT protein (Supplementary Fig. 20A).

We detected crosslinked PTBP1 RNPs in 4SU-RNA-treated cells expressing recombinant WT protein. Deletion of the N-terminus (PTBP1Δ1-59) had no effect but removal of RRM1 resulted in statistically significant loss of crosslinked PTBP1 RNPs (PTBP1Δ1-143, Fig. 5C and Supplementary Fig. 20C). After benchmarking with PTBP1, we tested whether mutagenesis of the RNA-binding interface on EXOC4 would affect RBP-RNA complex formation in cells. Since EXOC4 lacks a known RBD, we mutated residues flanking the evolutionarily conserved, RS-Cys site (C957) and detected a statistically significant decrease in EXOC4 RNP species compared with WT counterpart (EXOC4Δ947-967, Fig. 5C, Supplementary Fig. 20B, C). These data combined with direct identification of RNA bound to EXOC4 (Fig. 5B) further authenticates the RNA-binding activity of EXOC4.

Collectively, these mutagenesis experiments with known and unannotated RBPs provide additional evidence in support of PACCE discovery of functional cysteines involved in protein-RNA interactions in cells.

### Discussion

RNA-protein interactions[1-3] orchestrate complex networks of RNPs that regulate gene expression, translation, and epigenetic modulation[5,11]. While proteomic methods exist for identifying the composition of RBPs in cells[12,13], corresponding assays to assess RNA-binding activity of proteins are currently lacking. The ability to measure RBP activity states with binding site resolution is an important step towards elucidating the complete inventory of RBDs in the human proteome. Importantly, the use of chemical probes for RBP profiling is

needed to enable and streamline ligand discovery efforts through competitive activity-based protein profiling (ABPP) screening[46,66]. Here, we introduce PACCE as a chemoproteomic method to quantify RNA-binding activity of proteins directly in living cells.

A distinct feature of our approach compared with existing RIC methods is the use of a small molecule probe and not RNA to covalently bind, enrich, and identify RBPs in cells (Supplementary Fig. 14). We established CEPs as cysteine-reactive probes that do not require UV irradiation but covalently bind proteins via $S_N$Ar for ABPP profiling of RBPs and other purine-binding proteins in situ (Fig. 1). CEPs are readily integrated into modern chemoproteomic workflows[67] and do not require a RNA purification or extraction step to streamline rapid fluorescence gel-based or high-content mass spectrometry profiling of RBPs. The robust in situ activity of CEPs was important for maximizing RS-Cys coverage compared with in vitro probe labeling using CEPs and the broad-spectrum IA probe counterpart (Fig. 3A).

RNA binding sites have been mapped by LC-MS/MS analysis of RNA crosslinked peptides derived from RBPs[25]. While feasible, RNA-peptide adducts are heterogenous and require specialized proteomic workflows to deconvolute resulting data for increased resolution but at the cost of reduced sensitivity[1] (Supplementary Fig. 14). We developed an alternative 'footprinting' strategy for localizing protein-RNA interfaces that utilizes photoactivatable cellular RNA to crosslink and protect RNA binding regions on proteins from CEP probe labeling. The protected region(s), containing RS-Cys sites on proteins, was identified from the amino acid sequences of CEP-modified peptides displaying reduced LC-MS/MS abundance in the presence of RNA crosslinking competition. Importantly, our benchmarking experiments demonstrate (i) RNA specificity through loss of 4SU-RNA-dependent probe competition upon RNase treatment (Supplementary Fig. 12), and (ii) proximity of RS-Cys sites to the bound RNA across hundreds of RBP-RNA structures analyzed (Fig. 4).

We leveraged the standardized chemical probe format of CEPs and the broad diversity of RBPs amenable to RNA crosslinking competition to globally quantify RS-Cys on proteins with high resolution and sensitivity (~5500 candidate sites; Fig. 3 and Supplementary Data 4). The site specificity afforded by PACCE enabled domain enrichment analyses to discover RS-Cys that are prominent in both known and non-canonical RBDs (Fig. 3C and Supplementary Data 5). Notably, our method enabled identification of several functional domains including, for example, Q-motifs that can be further evaluated as RBDs in future studies (Supplementary Data 6).

The unbiased nature of PACCE was further showcased by discovery of RS-Cys across diverse protein classes that lacked prior RBP annotation. While several enriched protein functions were related to nucleotide recognition (e.g., nucleoside phosphate binding), a subset of proteins belonged to protein classes not obviously related to RNA binding (Fig. 3B). This latter class was best exemplified by EXOC4 and UIMC1, which were initially discovered by PACCE followed by verification of RNA-binding activity using orthogonal crosslinking and gel-shift assays[57] as well as direct validation of bound RNA by fPAR-CLIP[62] (Fig. 5B). As an additional control, we demonstrated that deletion of RNA-binding regions on RBPs identified by PACCE impaired formation of protein-RNA complexes in cells (Fig. 5C).

The discovery of RBP activity for UIMC1 is intriguing given the emerging roles of RBPs in the DNA damage response through direct repair or transcriptional and post-transcriptional control of gene expression[68-70]. Follow-up studies are needed to test whether the RNA-binding activity of UIMC1 is important for directing specificity of histone recognition at DNA DSBs. While the functional relevance of RNA binding activity for EXOC4 is less clear, we find it interesting that RBPs are implicated in loading RNA into extracellular vesicles[71,72].

Although the RS-Cys sites detected by PACCE are functionally enriched for RBDs, we are cognizant that sites meeting our competition threshold represent cysteine-containing regions with saturated

binding to cellular RNA. Crosslinking at higher energy (e.g., 254 nm) to increase native RNA-protein capture may further improve coverage of RS-Cys sites detected in proteomes[27]. These studies, however, will require additional controls to account for the general stress response using these more damaging photo-crosslinking conditions[1]. We note that our selection of 312 nm for UV irradiation in PACCE has the capacity for crosslinking native RNA to proteins in cells (Supplementary Fig. 9C and 10A). The versatility of 312 nm to crosslink native and 4SU-RNA was an important consideration given the reported 4SU incorporation rates in cells (1–4% of uridines[73]). Chemical crosslinking could increase efficiency for detecting RBP-RNA interactions but would require careful selection of controls to account for lower specificity due to protein-protein and protein-DNA crosslinks[74]. RS-Cys sites may not be located at the exact site of RNA crosslinking. We do not view this feature of PACCE as a limitation but rather a strength of the methodology to detect cysteine sites that are potentially involved in allosteric regulation of RNA recognition by proteins. In support of the latter, a recent report demonstrated the utility of using cysteine-reactive ligands to pharmacologically modulate RBP function[46].

We envision that PACCE will be highly complementary to RNA-focused global methods including PAR-CLIP[15] and SLAM-Seq[75] to bridge RNA sequence specificity with RBP activity in cells. Akin to other chemoproteomic methods, PACCE is well positioned to screen for covalent ligands using competitive ABPP methods[46,66] that can perturb RBPs with protein class- and binding site-selectivity across the human proteome[46]. While CEPs are broadly reactive probes of purine-binding proteins including RBPs (Fig. 1), additional structurally- and mechanistically-distinct chemotypes can further expand chemoproteomic investigations of this protein class. Importantly, the PACCE concept is versatile and can readily accept covalent probes that target more abundant residues found at RNA-protein interfaces[76] including, for example, lysines[77] and tyrosines[67].

In summary, we present a robust and high-content discovery platform for chemoproteomic discovery and quantitation of RNA binding activity on proteins directly in living cells. The ability for covalent binding to cysteines and potentially other ligandable residues in RBDs will help advance therapeutic discovery of this important class of 'undruggable' targets[41].

## Biological methods

### Cell culture
Cell lines were cultured with 5% $CO_2$ at 37 °C with manufacturer recommended media supplemented with 10% fetal bovine serum (FBS, U.S. Source, Omega Scientific) and 1% L-glutamine (Fisher Scientific): HEK293T, HeLa: DMEM; DM93, A549, Jurkat: RPMI. Cells were harvested for experimental use when they reached ~90% confluency. Plates were rinsed with cold PBS. Cells were scraped and washed (2X) with cold PBS with pelleting (400 x $g$, 3 min, 4 °C) and aspiration between washes. PBS was aspirated one final time before snap-freezing. Pellets were stored at −80 °C until further experimentation. HEK293T (CRL-3216), HeLa (CCL-2), A549 (CRM-CCL-185) and Jurkat (TIB-152) cells were purchased from the American Tissue Culture Collection (ATCC). DM93 cells were kindly provided by Dr. Seigler (Duke University Medical Center). Cell lines used for studies were not authenticated.

### SILAC cell culture
SILAC cells were cultured at 37 °C with 5% $CO_2$ in either 'light' or 'heavy' media supplemented with 10% dialyzed FBS (Omega Scientific), 1% L-glutamine (Fisher Scientific), and isotopically-labeled amino acids. HEK293T and HeLa cells were cultured in DMEM, while DM93, A549 and Jurkat cells were cultured in RPMI. Light media was supplemented with 100 µg/mL L-arginine and 100 µg/mL L-lysine. Heavy media was supplemented with 100 µg/mL [$^{13}C_6$$^{15}N_4$]L-arginine and 100 µg/mL [$^{13}C_6$$^{15}N_2$]L-lysine. The cells were cultured for 6 passages before use in

proteomics experiments. SILAC cells were harvest and preserved according to methods described in the cell culture section unless otherwise noted.

### Reactivity of CEP probes in situ
DM93 cells were cultured as described in the cell culture subsection. Cells were then treated with CEP probes (25 µM final, 50X stock in DMSO) for 4 h at 37 °C unless otherwise noted. The cells were then rinsed with cold PBS and collected via scraping and washed in cold PBS (2X). The pellet was reconstituted in PBS supplemented with cOmplete EDTA-free protease inhibitor tablets (EDTA-free) (Sigma-Aldrich, 11836170001) and sonicated (3 x 1 s pulses, 20% amplitude). Lysates were separated into soluble and membrane fractions using ultra-centrifugation (100,000 x $g$, 45 min, 4 °C). CEP-labeled proteomes were then prepared for click chemistry as described in the gel-based chemical proteomics section.

### Gel-based chemical proteomics
Gel-based click chemistry was performed as previously described[36] unless noted otherwise. Briefly, CEP-labeled samples were conjugated by copper-catalyzed azide-alkyne cycloaddition (CuAAC) to rhodamine-azide (1 µL of 1.25 mM stock; final concentration of 25 µM) using tris(2-carboxyethyl)phosphine (TCEP; 1 µL of fresh 50 mM stock in water; final concentration of 1 mM), tris[(1-benzyl-1H-1,2,3-triazol-4-yl)methyl]amine (TBTA, 3 µL of a 1.7 mM 4:1 t-butanol/DMSO stock, final concentration of 100 µM), and copper sulfate ($CuSO_4$, 1 µL of 50 mM stock, final concentration of 1 mM). The reaction was allowed to proceed for 1 h at room temperature. Once completed, the reaction was quenched with 17 µL of 4X SDS-PAGE loading buffer and beta-mercaptoethanol (βME). Quenched samples were analyzed by SDS-PAGE gel and in-gel fluorescence scanning on a Bio-Rad ChemiDoc MP Imaging System.

### Preparation of proteomes for SILAC LC-MS/MS chemical proteomics
CEP-labeled samples were prepared for LC-MS/MS analysis via CuAAC as described in the gel-based click chemistry section. Desthiobiotin-azide was supplemented for rhodamine-azide for purification. After incubation (1 h, room temperature), excess reagents were removed using chloroform/methanol extractions. The insoluble pellet was resuspended in 6 M urea, 25 mM ammonium bicarbonate (ambic). Proteins were reduced (DTT, 10 mM, 65 °C, 15 min), cooled (4 °C, 5 min) and alkylated (IAA, 40 mM, room temperature, dark, 30 min). Excess reagents were removed using chloroform/methanol extractions. Pellets were resuspended in 25 mM ambic (500 µL) and proteolytically digested (trypsin, 7.5 µg, 3 h, 37 °C). Samples were enriched using avidin beads and washed with PBS (3X). Bound peptides were eluted using 150 µL of 50% ACN + 0.1% formic acid (3X). Eluates were combined and centrifuged using mini Bio-Spin chromatography columns (Bio-Rad, 7326207) to remove additional avidin beads. Peptides were dried on a speed vac and reconstituted using 0.1% formic acid. Samples were stored at −80 °C until further analysis.

### Dataset comparisons
Iodoacetamide-desthiobiotin (DBIA) datasets from large-scale, cell-based screens (Kuljanin et al.[40]) were curated by Lai et al.[78] and used in the current study to obtain gene names that were converted to reviewed UniProt identifications (IDs) using mapping tools on UniProt (Human, Taxon ID: 9606). A total of 24,151 cysteine sites and 6,118 proteins from DBIA datasets were used for analysis and can be found in Supplementary Data 2. To approximate site position of other methods, the start and end position of the peptides was averaged. The average site was then truncated, generating the site (i.e., average down) for domain enrichment analysis. Overlap between CEP and reported datasets was calculated using a hypergeometric test[27].

### Nitrogenous base competition studies using gel-based chemical proteomics

HEK293T cells were grown according to the methods described in the cell culture section. Cell pellets were lysed in PBS supplemented with cOmplete EDTA-free protease inhibitor tablets (EDTA-free) (Sigma-Aldrich, 11836170001). Cellular fractions were separated with ultra-centrifugation (100,000 x *g*, 45 min, 4 °C). The soluble fraction was normalized to 2 mg/mL using the Bio-Rad DC protein assay. The lysate (48 μL) was mixed with CEP probe (AHL-Pu-1, 25 μM final, 50X stock in DMSO) and nitrogenous bases (adenine, uracil and cytosine: 500X stock in DMSO; purine: 5000X stock in DMSO). Nitrogenous bases were sonicated at 37 °C in a water bath sonicator to aid solubilization. Concentrations ranged from 25 μM to 2.5 mM for adenine, uracil, and cytosine, while purine concentrations ranged from 25 μM to 25 mM. Lysates were incubated at 37 °C for 30 min before labeling with the CEP probe. Next, click chemistry was performed and samples analyzed by gel-based chemical proteomics (*n* = 2 independent biological replicates).

### Nitrogenous base competition studies using LC-MS/MS chemical proteomics

HEK293T cells were cultured as described in the SILAC cell culture methods section. Soluble fractions were prepared as described in the nitrogenous base gel-based competition section. Protein concentrations were normalized to 2.3 mg/mL in 432 μL of PBS supplemented with cOmplete EDTA-free protease inhibitor tablets prior to the addition of either CEP probes or nitrogenous bases. Light SILAC cells were treated with AHL-Pu-1 (25 μM final), while heavy cells were co-treated with AHL-Pu-1 (25 μM final) and nitrogenous bases (adenine, uracil, cytosine: 2.5 mM final; purine 25 mM final) for 30 min at 37 °C. CEP-labeled samples were prepared as described in the preparation of proteomes for SILAC LC-MS/MS section. Competed sites were defined as probe-modified peptides that showed SR ≥ 5 with base competition and passed all other criteria listed in LC-MS/MS evaluation of peptides (*n* = 3 independent biological replicates).

### LC-MS/MS evaluation of peptides

Peptides were analyzed using nano-electrospray ionization-liquid chromatography-mass spectrometry (LC-MS/MS) on an Easy-nLC 1200 (Thermo Fisher) coupled to a Q Exactive Plus mass spectrometer (Thermo Fisher) utilizing a top 10 data-dependent acquisition mode (ddMS2)[36]. Reverse-phase LC was performed as follows: (A: 0.1% formic acid/H$_2$O; B: 80% ACN, 0.1% formic acid in H$_2$O): 0–1:48 min 1% B, 400 nL/min; 1:48–2:00 min 1% B, 300 nL/min; 2–90 min 16% B; 90–146 25% B; 146–147 min 95% B; 147–153 min 95% B; 153–154 min1% B; 154.0–154.1 min 1% B, 400 nL/min; 154.1–180 min 1% B, 400 nL/min.

### LC-MS/MS data analysis of CEP-modified peptides

Peptide identification from LC-MS/MS was accomplished using Byonic™ (Protein Metrics Inc.). Data were searched against the human protein database (UniProt, download date: 02/18/2016) with the following parameters: ≤2 missed cleavages, 10 ppm precursor mass tolerance, 20 ppm fragment mass tolerance, too high (narrow) "precursor isotope off by x", precursor and charge assignment computed from MS1, maximum of 1 precursor per MS2, 1% protein false discovery rate. Three variable (common) modifications were included: methionine oxidation (+15.9949 Da), cysteine carbamidomethylation (+57.021464 Da), and CEP-modified Cys (+604.2637). CEP probe modifications on amino acids of interest were included as a variable modification of 604.2637. Search results were filtered in R on a per site basis as previously reported[36]. The median SR for all cleavage patterns (i.e., ratios are calculated on a modified site basis, combining fully tryptic, half-tryptic and missed cleavages) was reported. Peptides highlighted in the main text and figures were manually validated[36]. Peptides with a SR >2 in PACCE conditions (4SU-RNA and 4SU- and

native-RNA; Supplementary Fig. 16) were analyzed with a Mann−Whitney *U*-test for significance compared to the SILAC mixing control (light and heavy proteomes added in a 1:1 ratio).

### Amino acid residue selectivity of CEP probes

DM93 SILAC cells were prepped and analyzed as described in SILAC cell culture section. Searches were accomplished using variable CEP probe modification (+604.2637) on the following nucleophilic amino acids: cysteine, aspartic acid, glutamic acid, histidine, lysine, methionine, asparagine, glutamine, arginine, serine, threonine, tryptophan, and tyrosine. A stricter Byonic™ score cutoff (≥600) was applied to further reduce false positive identifications.

### Gene Ontology (GO) analysis of proteins containing CEP probe modified sites

Combined CEP-modified protein lists were analyzed using either the Panther Classification System or GO Enrichment Analysis[38,39]. Charts generated using Panther used protein class enrichment on default settings: Fisher's exact test, Benjamini-Hochberg false discovery rate (FDR) < 0.05. Charts generated with GO Enrichment analysis utilized molecular function analysis on the following settings: binomial test, FDR correction.

### Domain enrichment analysis of CEP probe modified sites

Domain enrichment analysis of probe-modified sites was conducted as previously described[36]. *P*-values were calculated using a binomial test, which were then corrected with a 1% false-discovery rate (Benjamini-Hochberg correction).

### Identifying non-toxic conditions for 4SU metabolic incorporation into cellular RNA

Protocols were adapted from published reports[79]. HEK293T cells were cultured with varying concentrations of 4SU (100 and 500 μM) and metabolic incorporation allowed to proceed at short and longer time intervals (1–24 h). The 4SU concentrations were selected for testing based on literature reports[27]. 4SU incorporation in total RNA extracts from cells treated with varying conditions were determined by RNA dot blots as previously described[79]. The integrity of RNA extracts was assessed by the sharpness and lack of degradation of the distinct 28 S and 18 S ribosomal RNA bands using denaturing agarose gel analyses. At high concentrations (500 μM), substantial 4SU incorporation into cellular RNA was observed within 1 h and remained consistent across all time points tested. Higher 4SU incorporation into RNA was observed when used at lower concentrations (100 μM) for longer incubation times (16 and 24 h). Overt toxicity at these conditions was not observed as determined by cell viability measurements. Based on findings, optimal 4SU incorporation into cellular RNA was determined to be 100 μM for 16 h.

### Validating crosslinking of 4SU-RNA in cells

The xRNAx assay for purification of RNA-protein complexes was adapted from reference[22]. HEK293T cells (2.9 million cells/mL) were grown for 40 h (60% confluency). Cells were treated with 4SU (100 μM) for 16 h and crosslinked (1 J/cm$^2$) on ice. Media was removed and cells were rinsed with PBS before adding TRIzol. Once RNP complexes were purified, samples were reconstituted in RNase-free water and normalized to 75 ng in 40 mM Tris-HCl (pH 7.5). Samples were treated with either RNase A (Thermo Fisher Scientific, EN0531), DNase 1 (NEB, M0303L), or Proteinase k (Thermo Fisher Scientific, 25530049) at recommended concentrations for 1 h (37 °C). RNA loading dye (95% formamide, 0.125% SDS, and 0.1% EDTA) was added and samples were denatured for 5 min (65 °C). Samples were analyzed by 1% agarose gel electrophoresis. Ethidium bromide staining and imaging was performed on a Bio-Rad Che-miDoc MP Imaging System.

## Cell viability of CEP-treated cells

CEP-labeled DM93 cells were subjected to a WST-1 assay to assess cellular viability according to the manufacturer's protocol. Cells were plated in a 96-well dish. The tetrazolium salt was added to treated cells. The conversion of tetrazolium salt to formazan was measured after 30 min incubation at 37 °C using a BMG Labtech CLARIOstar plate reader. HEK293T cells were treated with 4SU in complete media (16 h, 100 µM, 100X stock in serum free media). Cells were then crosslinked (UV irradiation, 1 J/cm$^2$) on ice, scraped in cold PBS and centrifuged (400 x $g$, 3 min). The pellet was reconstituted in serum-free DMEM. Cells were then quantified using trypan blue according to the manufacturer's recommendations (Thermo Scientific) with a Countess™ II FL automated Cell Counter (Thermo Fisher). Statistics were calculated in Prism using a nonparametric, Kruskal-Wallis test of variance.

## PACCE in situ workflow for identification of RBPs

SILAC HEK293T cells were plated at 2.9 million cells per/mL and grown for 40 h (60% confluency). Cell media was replaced with 4SU-supplemented complete SILAC DMEM for 16 h. Afterwards, cells were washed with PBS and crosslinked at 312 nm (1 J/cm$^2$) on ice. Serum-free DMEM containing CEP (AHL-Pu-1, 25 µM final in DMSO) was added to cells after crosslinking. Probe labeling in cells was allowed to proceed for 1 h at 37 °C. RNase-free PBS was added and cells were harvested via scraping. Cells were then washed 2X times with RNase-free PBS (5 mL). The cell pellet was flash frozen and stored at −80 °C until further use.

Cells were lysed in PBS containing cOmplete EDTA-free protease inhibitor tablets (EDTA-free) (Sigma-Aldrich, 11836170001) by sonication on ice with an RNase free tip. Recombinant RNasin (rRNA, Promega, N2511) was added immediately after lysing according to the manufacturers protocol to protect crosslinked RNA. The cell lysates were then subjected to ultracentrifugation (100,000 x $g$, 45 min at 4 °C) to isolate the cytosolic fraction in the supernatant and the membrane fraction as a pellet. The membrane pellet was resuspended in a modified RIPA buffer (50 mM Tris-HCl, 1% NP-40 [Tergitol], 10% sodium deoxycholate, 10% SDS) with sonication. Protein concentrations were measured using the Bio-Rad DC protein assay. Soluble fractions were normalized to 2.3 mg/mL prior to chloroform/methanol extraction, while membrane samples were normalized to 3.3 mg/mL. Proteomes were prepared according to methods listed above. Custom peaks were added to the CEP modification in the Byonic™ search to account for probe fragments, including: +197.129 (d1), +240.1712 (d2), +425.2638 (d3), +453.2825 (d4). The PACCE-sensitive SR cutoff was lowered to ≥2 to account for low protein occupancy by RNA.

## RNase treatments

SILAC HEK293T cells were grown in 150 mm plates. Once at 80% confluency, heavy cells were treated with 100 µM 4SU for 16 h, resulting a 95% confluent plate. Cells were then washed with RNase free PBS, centrifuging at 1400 x $g$ for 3 min several times. Cells were then flash frozen in liquid N$_2$. Cells were lysed using a probe-tip sonicator (3 x 1 s pulse, 20% intensity) in 40 mM Tris-HCl (pH 7.5, 540 µL) with EDTA-free protease inhibitor. After sonication, 60 µL of 10x assay buffer (100 mM Tris-HCl, 25 mM MgCl$_2$, 5 mM CaCl$_2$, pH 7.6). Cell lysates were then combined, and an aliquot of each was taken to analyze concentrations. Aliquots were then split into each respective condition. RNase was added according to the manufacturer's recommendation. All samples without RNase were treated with recombinant RNasin according to the manufacturer's protocol. Samples were incubated at 37 °C for 1 h. When complete, concentrations were normalized to 2.3 mg/mL at 500 µL and transferred to a 12-well plate. Samples were crosslinked at 312 nm (1 J/cm$^2$) on ice. After, 432 µL were removed and treated with AHL-Pu-1 (25 µM final, 1 h, 37 °C). A desthiobiotin enrichment tag was then added using click-chemistry and the samples were processed using

chloroform/methanol extractions as described above. The MS parameters discussed above were used for analysis.

## Unenriched proteomics of cells grown under PACCE conditions

Cells were treated, lysed and proteomes processed as described in the PACCE workflow for identification of RBPs section. Soluble and membrane concentrations were then normalized to 2 mg/mL (100 µg L, 100 µg H) and subjected to a filter aided sample preparation (FASP) procedure with a 10 kDa cutoff filter. Samples were washed with PBS (300 µL) to remove probe (14,000 x $g$, 15 min). Protein was then reduced with DTT (5 mM final, 56 °C, 30 min). The mixture was then mixed with 200 µL urea/ammonium bicarbonate (UA, 8 M urea, 0.1 M Tris-HCl, pH 8.5) and centrifuged (14,000 x $g$ 15 min). An additional 200 µL was added and centrifuged again. An iodoacetamide solution (IAA, 0.05 M in UA, 100 µL) was added and incubated at room temperature for 20 min in the dark and then centrifuged at 14,000 x $g$ for 10 min. The samples were washed 3x times with 100 µL of UA and centrifuged (14,000 x $g$, 15 min). The filter was washed with ammonium bicarbonate (ABC, 100 µL, 0.05 M in H$_2$O) and centrifuged (14,000 x $g$, 10 min). The protein was digested with trypsin (1:100 trypsin to protein) on the membrane in ABC overnight at 37 °C. Once digested, an additional 40 µL of ABC was added to the unit and centrifuged (14,000 x $g$, 10 min) into a new tube. An additional 50 µL of 0.5 M NaCl was added to the top of the filter unit and centrifuged (14,000 x $g$, 10 min). The final peptide solution was acidified with acetic acid (5% final v/v).

C18- Stage tips (2 discs) were conditioned with 20 µL methanol followed by 20 µL 80% ACN, 0.1% acetic acid (buffer B), and then 20 µL water, 0.1% acetic acid (buffer A) (900 x $g$, 1 min). Loaded samples were washed with 20 µL of buffer A three times (900 x $g$, 1 min) and eluted with 20 µL of buffer B three times (900 x $g$, 1 min). Samples were dried down and stored at −80 °C. Reconstituted samples were analyzed using LC-MS/MS procedures described above.

Mass spectra were analyzed using Proteome Discoverer (PD, version 2.5) and searched using Byonic™ (v. 4.1.10) with a human protein database (UniProt 02/18/2016, 20,199 entries). The following search parameters were used: precursor and fragment ion mass tolerances ≤10 ppm and ≤ 50 ppm, respectively, signal to noise threshold ≥ 5, retention time shift ≤5 min, minimum peptide sequences per protein ≥ 1, peptide length ≥ 4, and a Byonic score threshold = 300. The protein false discovery rate was 0.01. One static modification was included: cysteine carbamidomethylation (+ 57.021464 Da). Several dynamic modifications (3 total) were included, including: methionine oxidation (+ 15.9949 Da, common 1), heavy lysine (+ 8.0142 Da, common 2), and heavy arginine (+ 10.0083 Da, common 2).

## RNA sequencing (RNA-Seq) of cells grown under PACCE conditions

Cells were grown and treated as described in the PACCE workflow for identification of RBPs section. RNA was then purified from cells using the PureLink™ RNA Mini Kit (Thermo Fisher, 12183018 A) and frozen (−80 °C). Samples were analyzed by Novogene (PE150, 6 G raw data). Briefly, messenger RNA (mRNA) was purified using poly-T magnetic beads and fragmented. Two strands were synthesized using random hexamer primers and dUTP or dTTP, respectively. The generated library was validated using Qubit and real-time PCR. Respectable libraries were pooled and sequenced using Illumina platforms.

Reads containing adapters, poly-N and low quality reads were removed, while Q20, Q30 and GC content were calculated on the remaining data. featureCounts (version 1.5.0-p3) was used to map to the Hisat2 (version 2.0.5) reference genome. Differential expression analysis was performed using DESeq2 (r package, 1.20.0). $P$-values were corrected using Benjamini and Hochberg's method for controlling the FDR ($P$-value ≤ 0.5). Read counts were adjusted using edgeR (3.22.5).

## Euclidean distance calculations between Cys residues and RNA

Crystal structures were downloaded from the Research Collaboratory for Structural Bioinformatics (RCSB) Protein Data Bank (PDB). Structures were filtered with the following criteria: protein co-crystallized with RNA, Homo sapiens, refinement resolution ≤3 angstroms, resulting in 270 structures. Structures with unnatural RNA (e.g., 5JS2) were manually removed. Structures were renumbered prior to processing (i.e., PDB numbers were matched to UniProt numbers) with PDBrenum[80]. Euclidean distances $(c^2 = x^2 + y^2 + z^2)$ were calculated between the sulfur atom of Cys (bio3d package, r) and any atom of the RNA, using the minimum calculated distance. Detected Cys sites were then matched to calculated distances. Structures that contained multiple chains produced slightly different distance measurements. All instances were used for analysis.

Domain information for all Cys residues found in co-crystal structures was obtained from UniProt. The complete list of HEK293T membrane data obtained from PACCE studies (RNA-sensitive and -insensitive sites) were then compared to the database. Sites classified as RBDs (i.e., KH, RRM. Helicase C-terminal, Helicase ATP and Double stranded RNA-binding) were defined as domains highlighted in Figs. 1D and 3C. Mean values were calculated for each corresponding group.

## Analysis of PACCE modified sites in intrinsically disordered regions

Intrinsically disordered regions from *Homo sapiens* were obtained from MobiDB predictions (v 4.1)[81].

## Alignment of PACCE-modified sites

Identified PACCE sites were aligned using Clustal defaults in Jalview (https://www.jalview.org/).

## PACCE in vitro workflow

SILAC HEK293T cells were treated with 4SU, crosslinked, flash frozen and stored at −80 °C as discussed above. Frozen pellets were thawed on ice. Cells were then reconstituted in PBS + protease inhibitor. The cells were lysed by passage through a 26 G needle (15 times). Recombinant RNasin was added according to the manufacture recommendation. The lysate was centrifuge at 100,000 x *g* for 45 min at 4 °C. Soluble and membrane were separated. The membrane samples were reconstituted in PBS + protease inhibitor using sequential passage through various needle sizes (18 G, 22 G, 26 G, respectively). Concentrations were normalized to 2.3 mg/mL (soluble) or 3.3 mg/mL (membrane) as describe in our in situ experiments. Lysates were treated with either 500 µM iodoacetamide-alkyne or 25 µM AHL-Pu-1 for 1 h at room temperature in the dark. The samples were then modified and processed using click chemistry and chloroform/methanol extractions as described above. Iodoacetamide-alkyne probe modifications on Cys residues were included as a variable modification of 509.2962. Custom peaks (as described above) were used for the identification of desthiobiotin fragments.

## Transient transfection of recombinant proteins

Recombinant protein production via transient transfection of HEK293T cells was performed according to published reports[36] with several modifications. Briefly, HEK293T cells were plated at 2.9 million cell/mL in complete DMEM and grown for 40 h to ~60% confluency. PTBP1 and EXOC4, were transiently transfected for 32 h followed by a 16-hr incubation with 100 µM 4SU prior to downstream processing. UIMC1 was transiently transfected for 47 h prior to the addition of 500 µM 4SU for 1 h. 4SU was reconstituted in serum-free DMEM and then added to complete DMEM for incubation. The following plasmid constructs (human proteins) were purchased from GenScript: pcDNA3.1-UIMC1-FLAG and pcDNA3.1-EXOC4-FLAG. Deletion mutant constructs were custom ordered from GenScript in a pcDNA3.1 + /C-

(K)-DYK plasmid backbone (PTBP1Δ1-59, PTBP1Δ1-143 and EXOC4Δ947-967).

## Fluorescence-based PAR-CLIP (fPAR-CLIP)

HEK cells were cultured as discussed above. Recombinant proteins were transiently transfected as described above. Cell pellets were stored at −80 °C until further processing. fPAR-CLIP experiments were carried out according to the protocol described previously[62]. Briefly, the cell pellets were resuspended in 3 volumes of IP buffer (20 mM Tris-HCl, pH 7.5, 150 mM NaCl, 2 mM EDTA, 1% NP40, 0.5 mM DTT), incubated with 0.1 U/µl RNase I (Thermo Fischer Scientific, AM2294) at 22 °C for 10 min, then 5 µl SUPERase•In/mL (Thermo Fisher Scientific, AM2694) was added to the cell extracts followed by centrifugation at 15,000 x *g* for 15 min at 4 °C. Immunoprecipitation (IP) was done by incubating the cleared extracts with anti-FLAG M2 magnetic beads (MilliporeSigma, M8823) (25 µL of beads per mL of cell lysate) for 90 min at 4 °C. Beads were washed three times with 1 mL IP buffer (without DTT) and resuspended in 2X bead volume IP buffer containing 1.5 U/µL RNase I at 22 °C for 10 min. Beads were washed twice with 1 mL IP buffer, twice with 1 mL high salt wash buffer (20 mM Tris-HCl, pH 7.5, 500 mM NaCl, 2 mM EDTA, 1% NP40) and twice with 1 mL CIP-PNK-Ligation wash buffer (50 mM Tris-HCl, pH 7.5, 10 mM MgCl$_2$). Dephosphorylation was carried out in 50 µL reaction mixture (5 µL cutsmart buffer, 2.5 µL Quick CIP, 2.5 µL SUPERase•In, 40 µL nuclease free H2O) (Quick CIP, NEB, M0525S) at 37 °C for 10 min with shaking. Next beads were washed three times with 1 mL CIP-PNK-Ligation wash buffer. To fluorescently label the ribonucleoprotein complex to allow for visualization, on beads 3' fluorescent adapter ligation was performed in 50 µ reaction mixture (0.5 µl 50 µM fluorescent 3' adapter, 5 µL 10× T4 RNA ligase reaction buffer, 15 µl 50% aqueous PEG-8000, 2.5 µl T4 Rnl2(1–249)K227Q ligase, 2.5 µl SUPERase•In, 24.5 µl nuclease free H2O) (Fluorescent barcoded 3'adapter: 5'-rAppNNTGACTGTG-GAATTCTCGGGT(fl)GCCAAGG-fl*, T4 Rnl2(1-249)K227Q ligase, NEB, M0351) at 4 °C for overnight with gentle agitation. Next, beads were washed twice with 1 mL high salt wash buffer and resuspended in 60 µl 2x loading buffer (NuPAGE LDS Sample Buffer +DTT, Thermo Fisher Scientific, NP0008) and incubated for 5 min at 95 °C. The supernatants were separated on NuPAGE 4-12% Bis-Tris Midi Protein gel (Thermo Fischer Scientific, WG1401BOX) alongside 20 µl of 1/10 dilution of PageRuler Plus Prestained Protein Ladder (Thermo Fisher Scientific, 26620). After electrophoresis, gel was placed in clear plastic sheet protector and scanned for AF647 using a GE Typhoon 9500 scanner to visualize bands corresponding to the RBP-fluorescent 3'adapter ligation products.

## Western blots of RNA-sensitive proteins

Western blot analysis of recombinant protein and mutants was performed as previously described[36] with several modifications. Cells overexpressing each respective plasmid were crosslinked (1 J/cm$^2$) on ice. Cells were then scraped in RNase-free PBS and pelleted (1,400 x *g*, 3 min). Cells were reconstituted in IP buffer and lysed using a 26-gauge syringe. Samples without RNase treatment were spiked with RNasin and incubated on ice. The RNase treated sample was incubated with RNase A and RNase I as described by the manufacturer for 10 min at room temperature. Lysates were then cleared at 15,000 x *g* for 15 min at 4 °C. Samples were loaded onto the gel with Laemmli loading dye and heated for 1 min at 95 °C. The following primary antibody was used: anti-FLAG antibody (Sigma-Aldrich, F7425-2MG, 1:1000). The following primary antibody was used as a loading control: Goat anti-GAPDH IgG (Cell signaling Technology, 2118 S, 1:1,000). Appropriate secondary antibodies were purchased from Thermo Fisher (1:10,000).

## Polynucleotide kinase (PNK) assay

PNK addition of γ-32P rATP was conducted as previously described with various modifications as highlighted below[63]. Proteins were

recombinantly overexpressed, 4SU treated, and crosslinked as described above. Cell pellets were flash frozen and stored at −80 °C until use. Cells were lysed in lysis buffer (100 mM NaCl; 50 mM Tris-HCl pH 7.5; 0.1% SDS; 1 mM MgCl$_2$; 0.1 mM CaCl$_2$; 1% NP40; 0.5% sodium deoxycholate; protease inhibitors (Roche, 11836170001) via 10 passages through a 26 g needle on ice). Lysates were then cleared at 15,000 x $g$ for 15 min at 4 °C. The supernatant was treated with various concentrations of RNase A (8 ng/µL, 2 ng/µL and 0.5 ng/µL; Thermo Fisher, EN0531) and 2 U/mL DNase (New England Biolabs, M0303L) for 15 min at 37 °C. IP was conducted at 4 °C for 2 h. The samples were then washed 3 times with lysis buffer followed by two additional washed with PNK buffer (50 mM NaCl; 50 mM Tris-HCl pH 7.5; 10 mM MgCl$_2$; 0.5% NP-40; protease inhibitors (11873580001, Roche)). Magnetic columns were then capped, and samples were incubated in 0.1 µCi/µl [γ-32P] rATP, 1 U/µl T4 PNK (NEB), 1 mM DTT and labeled for 15 min at 37 °C. Samples were eluted with hot Laemmli before separation on an SDS-PAGE gel and transferred to a PVDF membrane. Membranes were then exposed to an autoradiographic film, and the developed films were scanned using Amersham ImageQuant800™ imaging systems.

### Functional validation of RNA-sensitive Cys residues
Cell pellets were reconstituted in 40 mM Tris-HCl (pH 7.5) with protease inhibitor. The slurry was then probe tip sonicated on ice with an RNase free tip. Recombinant RNasin (rRNA, Promega, N2511) was added immediately after lysing according to the manufacturers protocol to protect crosslinked RNA. Proteins were separated via SDS-PAGE. Proteins of interest were visualized using anti-FLAG antibodies. EXOC4 data was analyzed using a quantile-quantile (QQ) plot to assess normality followed by a ratio paired $t$-test (paired, parametric) to determine significance. PTBP1 was also tested for normality. The data were analyzed using an RM one-way ANOVA without corrections followed by a Tukey's post hoc test for multiple comparisons.

### Reporting summary
Further information on research design is available in the Nature Portfolio Reporting Summary linked to this article.

## Data availability
The data supporting the findings of this study are available from the corresponding authors upon reasonable request. Proteomics data have been deposited at ProteomeXchange via the PRIDE database (http://www.proteomexchange.org) and are publicly available under accession numbers PXD044625 and https://doi.org/10.6019/PXD044625. RNA-seq data are available on NCBI Gene Expression Omnibus, under accession number GSE240318. The crystallographic data supporting this work are deposited at the Cambridge Crystallographic Data Centre (CCDC) under the CCDC deposition number 2272256. These data can be obtained free of charge from CCDC via www.ccdc.cam.ac.uk/structures. Source data are provided with this paper.

## Code availability
The custom code used in the manuscript is explained in detail in the methods section so that it can be readily re-created by other groups. The code can be provided upon request.

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

## Acknowledgements

We thank B. Pachaiyappan for initial help in the synthesis of purine probes. We thank M. Shin for initial assistance with proteomic analyses of purine probes. We thank M. Ross and A. Borne for assistance with data analysis. We thank T. Ware for helpful discussions on the project. We thank D. Dickie for assistance with small molecule crystallography studies. We thank all members of the Hsu lab for helpful discussions and review of the manuscript. This work was supported by the National Institutes of Health grant nos. GM136615 (A.J.H.), GM007055 (J.W.B.), GM148379 (M.C.L.), GM136900 (T.E.H.), GM144472 (K.-L.H.), DA043571 (K.-L.H.), AI169412 (K.-L.H.), University of Virginia Cancer Center (NCI Cancer Center Support Grant No. 5P30CA044579-27 to K.-L.H.), the Owens Family Foundation (K.-L.H.), the Intramural Research Program of the National Institute for Arthritis and Musculoskeletal and Skin Diseases (X.W. and M.H.), the Robbins Family MRA Young Investigator Award from the Melanoma Research Alliance (https://doi.org/10.48050/pc.gr.80540 to K.-L.H.), the Mark Foundation for Cancer Research (Emerging Leader Award to K.-L.H.), and a Recruitment of Rising Stars Award from CPRIT (RR220063 to K.-L.H.).

## Author contributions

Conceptualization, A.J.H. and K.-L.H.; Methodology, A.J.H., J.W.B. and K.-L.H.; Investigation A.J.H., J.W.B., M.W.F., X.W., M.H., M.C.L., D.M.L.; Synthesis, A.H.L.; Data analysis and validation, A.J.H., D.L.B., X.W., M.C.L., T.E.H., M.H. and K.-L.H.; Distance calculation analysis, A.J.H. and D.L.B.; Writing—original draft, A.J.H., A.H.L., X.W., M.H. and K.-L.H.; Writing—review and editing, A.J.H. and K.-L.H.; Funding acquisition, A.J.H., J.W.B., X.W., M.H. and K.-L.H.; Supervision, K.-L.H.

## Competing interests

The authors declare no competing interests.
