## [Peer Review File · Nature Communications]

Reviewers' Comments:

Reviewer #2:

Remarks to the Author:

Reviewer's comments on the revised manuscript by Heindel et al. "Chemoproteomic capture of RNA binding activity in living cells"

First at all I would like to apologize for the very much delayed review of the revised version of the manuscript.

It is my opinion that the authors have significantly improved their manuscript. I am particularly happy that the comparison of their CEP probe with iodoacetamide is now more reliable. Furthermore, the authors have addressed my other points in a sufficient manner.

Therefore, I am happy to state that the revised version of the manuscript should be published in Nature Communications.

Reviewer #3:

Remarks to the Author:

The authors describe the development of a method termed Photo-Activatable-Competition and Chemoproteomic Enrichment (PACCE) for the global quantification of protein-RNA contacts in cells. As described in detail below, I am not convinced about the performance of PACCE. I divided my comments in three main conceptual areas.

1) Specificity of the CEP probes towards RBPs

The authors compare the activity of CEP and of general cysteine-reactive probes (e.g. DBIA). These analyses in my opinion are instrumental to assess the performance of the CEP probes. Unfortunately, I think that the overall conclusion of these comparisons, as well as of other presented data (see below), is that the proposed CEP probes are a good tool to target cysteines in cellular proteins (Figure S8), but that are not an excellent choice to study RBPs. While the number of RBPs identified using the CEP probes might be acceptable, I am not convinced about the specificity of the probes. Note that among CEP-target regions, there are 5232 hits mapping to proteins involved in "protein binding" and just 1102 mapping to proteins with "RNA binding" function (and among these two groups, the cluster "protein binding" displays in fact a much lower FDR) (Table S3, GO molecular function). Also note that the percentage of identified hits in the group "RNA metabolism protein" displays only a very subtle increase from 12% to 14-15% in CEP-based studies relative to DBIA-based studies (Figure S9B). Furthermore, DBIA-based datasets show a much stronger enrichment of proteins bearing bonafide RNA-binding domains (as helicase ATP-binding and helicase C-terminal) relative to CEP-derived datasets.

I think that the low specificity of CEP probes towards RBPs is not surprising considering that the employed probe(s) just modestly resemble the regions in cellular RNAs recognized by RBPs (that are several nucleotides long and that can result from RNA folding into specific secondary structures). In addition, the amino acid preferentially targeted by the CEP probes, cysteine, is not common among RNA-binding domains (it is in fact negatively represented) (PMID: 16790841).

2) Low efficiency of UV crosslinking of protein-RNA interactions in PACCE

Regarding the PACCE approach, besides the aforementioned considerations, there is a conceptual problem: UV crosslinking is inefficient for numerous RBPs. Available data show that as little as 5% of a

given RBP can be crosslinked to RNA (PMID: 33332543; PMID: 31586594). That means that the reduction upon irradiation of the pool of cysteines "free" to react with the CEP probes is expected to be very subtle for these proteins, which would compromise the detection of RNA-interacting cysteines. If the authors know that the crosslinking strategy they apply leads to much higher average efficiencies (e.g. due to the UV wavelength, use of 4SU, etc.), it would be useful to state it and show the supporting data. Otherwise, and for the exposed reason, I would assume that the utility of PACCE is rather narrow. Note that the results presented in Figure S17 are of limited value, given the fact that RNase treatment is expected to cause massive changes in the crowding of the lysates as well as in the solubility of the proteins (PMID: 32945072). This could influence cysteine reactivity indirectly rather than by means of a competition between RNA and the probe, as suggested. The use of +/- UV irradiation in the absence of any RNase treatment might be a better option. Furthermore, these analyses should be done in cells, given that the authors detect 7 times more "RNA-sensitive" cysteines with in situ PACCE relative to the in vitro variant, and that that is the focus of the rest of the paper.

3) Lack of proper validation of PACCE

It would be essential to validate further and properly the capacity of PACCE to detect RNA-binding domain/residues, which ideally should involve multiple proteins. I find that the supposed validation currently presented in the paper (Figure 5) is NOT conclusive. The authors claim: "...we observed a shift in molecular weight upon UV irradiation of 4SU-labeled, EXOC4-expressing HEK293T cells that was not observed in the absence of UV irradiation and reduced with RNase treatment." But I don't think this is true; albeit weaker, the same boxed bands are still visible in non-irradiated samples, and it is not clear why. In addition, the RNase-response is extremely subtle and does not present the expected pattern (see PTBP1 in Fig 5B). Analogous suboptimal results are presented for the other chosen candidate, UIMC1. In Figure 5C, note the occurrence of multiple bands in each lane, from which the authors select those that match the expected MW to claim RNA association. Given the lack of western blot analysis and proper controls (e.g. RNase titration), it is not possible to judge whether these bands correspond or not to the alleged proteins (or for example to contaminating RBPs), and it is not possible to judge neither whether the signal actually comes from labelled RNA (or, e.g., from a direct "unspecific" interaction of the protein with the fluorescent oligoribonucleotide; a type of interaction I have seen happening in gel).

Reviewer #4:

Remarks to the Author:

Heindel et al developed a chemoproteomic strategy to identify RNA-binding proteins and sites in living cells. The method is based on an electrophilic purine analogue probe that shows extensive labeling of cysteines in proteomes and the authors applied it in a competitive profiling manner when SILAC cells were labeled with 4SU and irradiated with UV to induce crosslinking between RNA and proteins. The rationale is that the 4SU crosslinking at the protein-RNA interface would interfere with the purine probe labeling and a cysteine site that is quantified with a high SILAC ratio would be a potential target site proximal to the RNA crosslinked site from which an RNA-binding protein can be discovered. They did thorough analyses on the list of proteins and cysteine sites from the profiling and biochemically characterized some targets that were previously unknown with RNA-binding activity. They finally demonstrated that a novel target EXOC4 has a RNA-binding activity and plays a role in RNP formation. Overall, I think the technical novelty of this method is quite clear, which is distinguishable with other profiling methods of RNA-binding proteins. Although the competitive chemoproteomic profiling of cysteines have been extensively reported before (which by the way I think the authors should reference), this is the first report to my knowledge of being applied for RNA-binding proteins. The profiling efforts have resulted in a couple of datasets that should be of broad interest to chemical biology and RNA biology community – namely, the overall proteome reactivity profile of the purine probe and the 4SU competitive profiling data of this probe for hunting RNA-binding proteins. However, my major concern is the data presentation, especially the graphic demonstration and the writing styles, which have already been raised by other reviewers. I don't think the manuscript has been

improved in terms of these aspects and I still find it challenging to read through. Here are some specific points:

1) Figure 1 and 2 are with too many texts and should be re-drawn with attractive graphics/cartoons/ for clarity.

2) The rest of figures are also too crowded and each subfigure should have enough margins left from its surrounding subfigures and with proper font sizes (some of which are too large to be confusing, e.g., figure 5B and 5C)

3) The writings made it hard to follow the overall logic flow of this manuscript. The result sections are overwhelmed with too many technical details such as list of cell line names, percentage values, as well as abbreviation names coined by authors themselves (e.g, SR>2, 1:1 control). The authors should make an effort to polish the writing to convey main messages in each section and refer to the key figures to help support such messages.

4) Based on the current frame, I suspect that the authors first made the purine probe and hope it can achieve the profiling of cysteines at the RNA-protein interfaces alone. Then the competitive labeling was introduced to narrow down those cysteines that are perturbed by 4SU photocrosslinking. If so, my suggestion is to establish the purine probe just as a generic cysteine-reactive probe for chemoproteomic profiling in living cells and then established the whole "PACCE" workflow to focus on the RNA-binding proteins. As also suggested by other reviewers, it needs more evidences to support the purine probe as the cysteine-reactive probes of the RNA-interactomes as there are clearly other types proteins being pulled out by the purine probe. Only when it is used together with the photoactivatable 4SU competition, the probe maximized its impact in capturing key cysteines at the RNA-protein interfaces.

5) Although there is no hard limit, 30 are abnormally large for the number of SI figures and it will clearly hurt the overall readability of this manuscript. The authors are suggested to consolidate some of these data.

REVIEWER COMMENTS

Reviewer #2 (Remarks to the Author):

Reviewer's comments on the revised manuscript by Heindel et al. "Chemoproteomic capture of RNA binding activity in living cells"

First at all I would like to apologize for the very much delayed review of the revised version of the manuscript.

It is my opinion that the authors have significantly improved their manuscript. I am particularly happy that the comparison of their CEP probe with iodoacetamide is now more reliable. Furthermore, the authors have addressed my other points in a sufficient manner.

Therefore, I am happy to state that the revised version of the manuscript should be published in Nature Communications.

We thank the reviewer for such a positive review!

Reviewer #3 (Remarks to the Author):

The authors describe the development of a method termed Photo-Activatable-Competition and Chemoproteomic Enrichment (PACCE) for the global quantification of protein-RNA contacts in cells. As described in detail below, I am not convinced about the performance of PACCE. I divided my comments in three main conceptual areas.

1) Specificity of the CEP probes towards RBPs

The authors compare the activity of CEP and of general cysteine-reactive probes (e.g. DBIA). These analyses in my opinion are instrumental to assess the performance of the CEP probes. Unfortunately, I think that the overall conclusion of these comparisons, as well as of other presented data (see below), is that the proposed CEP probes are a good tool to target cysteines in cellular proteins (Figure S8), but that are not an excellent choice to study RBPs. While the number of RBPs identified using the CEP probes might be acceptable, I am not convinced about the specificity of the probes. Note that among CEP-target regions, there are 5232 hits mapping to proteins involved in "protein binding" and just 1102 mapping to proteins with "RNA binding" function (and among these two groups, the cluster "protein binding" displays in fact a much lower FDR) (Table S3, GO molecular function).

We should clarify that the 5232 hits to "protein binding" and 1102 hits to "RNA binding" proteins correspond to DBIA and not the CEP datasets. If we directly compare the "RNA binding" GO function (GO:0003723) between the probes, we find CEP shows further enrichment compared with DBIA (3-fold for CEP (found in Table S3, 'ALLMF' tab) versus 2-fold enrichment for DBIA (found in Table S3, 'DBIAMF' tab).

Also note that the percentage of identified hits in the group "RNA metabolism protein" displays only a very subtle increase from 12% to 14-15% in CEP-based studies relative to DBIA-based studies (Figure S9B).

We should re-emphasize that the strength of CEPs and PACCE is the ability to perform proteomic investigation of RBPs in live cells, which we (Figure 3A) and others in the RBP field (a comprehensive review provided in PMID: 29339797) have shown to be critical for capturing RBP activity. The ability of

CEPs to capture RNA interaction sites on proteins is substantially improved in live cells compared with in vitro labeling using existing cysteine-reactive probes (3-fold increase for CEP vs DBIA, Figure 3A).

Furthermore, DBIA-based datasets show a much stronger enrichment of proteins bearing bonafide RNA-binding domains (as helicase ATP-binding and helicase C-terminal) relative to CEP-derived datasets.

While this is true, we should point out that the CEP probe shows stronger enrichment of proteins containing the RRM, KH, and double-stranded (dsRNA) RNA-binding domains (Figure 1D). RRM, KH and dsRNA domains are well-validated and important RNA-binding domains with numerous reports in literature (PMID 15853797, 36689472, 8405383, 11743725, 15573138, 22918483).

This comparison also highlights an important point: despite a nearly 4-fold decrease in cysteine site coverage using CEP (6214 vs 24151 cysteine sites detected by CEP vs DBIA, respectively; Figure 1D), the CEP probe can enrich for RNA-binding domains to a comparable and in some cases enhanced degree compared with the state of art cysteine-reactive probe DBIA.

I think that the low specificity of CEP probes towards RBPs is not surprising considering that the employed probe(s) just modestly resemble the regions in cellular RNAs recognized by RBPs (that are several nucleotides long and that can result from RNA folding into specific secondary structures). In addition, the amino acid preferentially targeted by the CEP probes, cysteine, is not common among RNA-binding domains (it is in fact negatively represented) (PMID: 16790841).

Cysteines are underrepresented in canonical RNA binding domains but we believe this could be a strength for developing small molecule modulators of this protein class. A recent report demonstrated the utility of developing cysteine-reactive ligands to target the RBP NONO for altering the transcriptomes of cancer cells via ligand induced stabilization of RBP-RNA interactions (PMID 36864190). These findings provide important proof of concept that cysteines found in allosteric regions of RBPs are ligandable (and potentially druggable) sites for pharmacological modulation. Our initial evaluation of cysteine site sensitivity to RNA crosslinking in relation to RNA-cysteine distance (Figure 4C) provides a resource for enabling future investigation of direct and allosteric modulators of RBP function.

2) Low efficiency of UV crosslinking of protein-RNA interactions in PACCE

Regarding the PACCE approach, besides the aforementioned considerations, there is a conceptual problem: UV crosslinking is inefficient for numerous RBPs. Available data show that as little as 5% of a given RBP can be crosslinked to RNA (PMID: 33332543; PMID: 31586594). That means that the reduction upon irradiation of the pool of cysteines “free” to react with the CEP probes is expected to be very subtle for these proteins, which would compromise the detection of RNA-interacting cysteines.

4SU is a well-established methodology used in the RBP and RNA community for the past ~60 years (PMID 5321899) and a search on PubMed will reveal hundreds of published reports using 4SU crosslinking for RBP profiling. PACCE could be improved if a more efficient crosslinking version of 4SU could be established in the field. However, 4SU remains the state of the art and development of a new crosslinkable RNA nucleoside is beyond the scope of the current manuscript.

If the authors know that the crosslinking strategy they apply leads to much higher average efficiencies (e.g. due to the UV wavelength, use of 4SU, etc.), it would be useful to state it and show the supporting data. Otherwise, and for the exposed reason, I would assume that the utility of PACCE is rather narrow.

We considered several avenues for RNA crosslinking when developing PACCE and chose UV crosslinking at 312 nm using 4SU based on reports demonstrating these conditions can enhance specificity for protein-RNA crosslinks with minimal loss to overall proteome sensitivity (PMID 27768875 and 24141703). As the reviewer pointed out, formaldehyde has been used for investigating RBP-RNA interactions and while more efficient compared to UV, this chemical crosslinking technique can introduce additional artifacts through protein-protein (PMID 32561732, 17593931) and protein-DNA crosslinks (PMID 25101823, 10606672, 24848408), which would further confound our competition studies. There are ongoing discussions in literature regarding the advantages and disadvantages of using UV compared with formaldehyde for protein-RNA crosslinking, but a general consensus is that UV is more specific (PMID 30804549). Given that we are developing a new chemoproteomic methodology for RBP profiling, we prioritized specificity for our proof-of-concept studies. We should point out that PACCE is amenable to chemical crosslinking, and we can explore formaldehyde-mediated RBP-RNA interactions in future studies.

To further clarify our rationale for choosing 4SU-RNA crosslinking at 312 nm, we updated the results section in the revised manuscript as follows:

“We chose to incorporate 4SU for UV-mediated crosslinking of cellular RNA to protein at 312 nm because of (i) higher specificity (e.g., DNA-protein crosslinks and single-strand breaks are ~1000-fold less at 312 compared with 254 nm⁴²) without compromising proteomic sensitivity^{27,43}, (ii) less damage to cells^{1,27}, and (iii) a UV wavelength closer to the optimum extinction coefficient of 4SU⁴⁴ (Figure S4 and S9)”

We also updated the limitations/future directions section of the discussion as follows:

“The versatility of 312 nm to crosslink native and 4SU-RNA was an important consideration given the reported 4SU incorporation rates in cells (1-4% of uridines⁷³). Chemical crosslinking could increase efficiency for detecting RBP-RNA interactions but would require careful selection of controls to account for lower specificity due to protein-protein and protein-DNA crosslinks⁷⁴.”

Note that the results presented in Figure S17 are of limited value, given the fact that RNase treatment is expected to cause massive changes in the crowding of the lysates as well as in the solubility of the proteins (PMID: 32945072). This could influence cysteine reactivity indirectly rather than by means of a competition between RNA and the probe, as suggested. The use of +/- UV irradiation in the absence of any RNase treatment might be a better option. Furthermore, these analyses should be done in cells, given that the authors detect 7 times more “RNA-sensitive” cysteines with in situ PACCE relative to the in vitro variant, and that that is the focus of the rest of the paper.

We do not believe indirect, proteome-wide effects are occurring in our RNase treatments. If RNase treatments were causing a massive, non-specific alteration in proteomes, the SILAC ratios of peptides (both probe-modified and non-modified peptides) would be heavily skewed (SR >>2 and/or SR <<2) in (+)RNase vs. (-)RNase conditions. We do not observe this type of effect. In fact, we determined the median SILAC ratio of all peptides detected in our RNase treated datasets, averaged this value across 3 biological replicates and found this value to be ~1.2.

The in situ RNase treatment is a potentially interesting experiment but will be confounded by indirect effects because of global re-localization of RBPs from the cytoplasm to the nucleus from RNase

activation in cells (e.g., PMID 36318584). Thus, introduction of an active RNase to cells could influence cysteine reactivity indirectly because of the accessibility of RBPs to probe labeling and not strictly from competition between RNA and CEP probe.

3) Lack of proper validation of PACCE

It would be essential to validate further and properly the capacity of PACCE to detect RNA-binding domain/residues, which ideally should involve multiple proteins. I find that the supposed validation currently presented in the paper (Figure 5) is NOT conclusive. The authors claim: "...we observed a shift in molecular weight upon UV irradiation of 4SU-labeled, EXOC4-expressing HEK293T cells that was not observed in the absence of UV irradiation and reduced with RNase treatment." But I don't think this is true; albeit weaker, the same boxed bands are still visible in non-irradiated samples, and it is not clear why.

We suspect crosslinking from ambient light is producing the higher molecular weight species observed for recombinant EXOC4-expressing HEK293T cells in the (-)UV condition. This behavior (i.e., cross-linked RNP signals in the absence of UV) has been reported for several known RBPs in literature reports (e.g., Figure 1F and G from PMID 30824702).

We have modified the results section as follows to clarify this point:

"Akin to the migration behavior of a known RBP (Polypyrimidine tract-binding protein 1 or PTBP1), we observed a shift in molecular weight upon UV irradiation of 4SU-labeled, EXOC4-expressing HEK293T cells that was muted in the absence of UV irradiation and reduced with RNase treatment (S18A and B). The higher molecular weight signals observed without UV irradiation are likely due to crosslinking from ambient light as previously reported⁵⁷."

In addition, the RNase-response is extremely subtle and does not present the expected pattern (see PTBP1 in Fig 5B). Analogous suboptimal results are presented for the other chosen candidate, UIMC1.

A reduction but not complete ablation of RNA-binding signal with RNase treatment of RBPs is not uncommon in literature. For example, in the PTex report (PMID 30824702), the cross-linked RNP signals for known (PTBP1, FUS; Figure 1F and G) and candidate RBPs (ABCF2, CCT7; Figure 5H and I) are not completely ablated with RNase treatments. In some cases, only a subtle decrease was observed.

We performed additional control experiments to verify protein-RNA interactions by the polynucleotide kinase (PNK) assay as previously reported (PMID: 30773316). Cellular lysates from UV (312 nm) irradiated, 4SU-RNA-treated HEK293T cells recombinantly expressing PTBP1, EXOC4 or UIMC1 were subjected to RNaseA at increasing concentrations. Afterwards, proteins were immunoprecipitated followed by radioactive labeling (³²P) of RNA 5' ends with T4 polynucleotide kinase and imaging of autoradiographic film. Western blotting was performed to confirm protein expression across samples.

As shown in the Figure below, we observed increased higher molecular weight radiolabeled bands upon UV irradiation of HEK293T cells overexpressing protein and this smeared signal corresponding to RNP complexes was reduced in a RNaseA-concentration dependent manner. Note that the protein loading for (-)RNase treatment sample of UIMC1 was higher than the other samples but RNaseA concentration dependence was still apparent when comparing 0.5 and 2 ng/μL RNaseA-treated samples.

Figure S18F. Control experiments to verify protein-RNA interactions by the polynucleotide kinase (PNK) assay.

We have included this new data as Figure S18F in the revised manuscript. The results section has also been updated as follows:

“We further verified protein-RNA interactions by the polynucleotide kinase (PNK) assay⁶³. Cellular lysates from UV (312 nm) irradiated, 4SU-RNA-treated HEK293T cells recombinantly expressing PTBP1, EXOC4 or UIMC1 were subjected to RNaseA at increasing concentrations. Afterwards, proteins were immunoprecipitated followed by radioactive labeling (³²P) of RNA 5' ends with T4 PNK. As shown in the Figure S18F, we observed increased higher molecular weight radiolabeled bands upon UV irradiation of HEK293T cells overexpressing RBP and this expected ‘smeared’ signal corresponding to RNP complexes was reduced in a RNaseA-concentration dependent manner.”

We recognize that the RNase response for UIMC1 is more subtle compared with PTBP1 and EXOC4 and thus performed additional biological replicates to quantify the reduced RNP signals for UIMC1. The new data show significant reductions in UIMC1 RNP signal upon RNase treatments and are included in revised Figure S18C and attached below:

Figure S18C. Updated western blot analysis showing RNP formation upon UV crosslinking in HEK293T cells expressing recombinant protein UIMC1 that was significantly reduced with RNase treatment.

In Figure 5C, note the occurrence of multiple bands in each lane, from which the authors select those that match the expected MW to claim RNA association. Given the lack of western blot analysis and proper controls (e.g. RNase titration), it is not possible to judge whether these bands correspond or not to the alleged proteins (or for example to contaminating RBPs), and it is not possible to judge neither whether the signal actually comes from labelled RNA (or, e.g., from a direct “unspecific” interaction of the protein with the fluorescent oligoribonucleotide; a type of interaction I have seen happening in gel).

We repeated the fPAR-CLIP experiment and included a mock-transfected (empty expression plasmid) control. As you can see in the revised Figure 5B, all background fluorescent bands (which may derive from “unspecific” interactions of proteins with fluorescent oligoribonucleotides) are present in the mock control, which is clearly differentiated from the specific fluorescent bands from crosslinking FLAG-PTBP1, FLAG-EXOC4 and FLAG-UIMC1 RNPs. Please note that for fPAR-CLIP the migration of crosslinked, adapter ligated RNPs is expected to run ~20 kDa above the expected migration of non-crosslinked RNPs (PMID: 33503264), as seen in the matching Western Blot in Figure 5B. Please also note that we previously found that at the 4SU concentrations used for PAR-CLIP and fPAR-CLIP (100 μ M, 16 hrs pulse) we expect a substitution of 1 in 40 Us with 4SU (PMID: 20371350). This in turn means that only a small fraction of the RBP bound transcriptome will crosslink and thus, the 20 kDa shift due to adapter ligation will not be reflected in the Western Blots.

Figure 5B. Updated fPAR-CLIP analyses with matching western blots.

We performed RNase treatments at high and low concentrations in the fPAR-CLIP studies as suggested by the reviewer (see below). Similar to observations with other RBPs (PMID: 33503264), the migration pattern of crosslinked and adapter-ligated RNPs did not change dramatically. As fPAR-CLIP uses high RNase concentrations for stringent footprinting, we typically only observe dramatic effects for RNase titrations when dealing with large complexes that contain multiple RBPs.

fPAR-CLIP studies performed with high (1.5 U/μL) and low (0.15 U/μL) RNase treatments.

We direct the reviewer to our PNK ³²P assay result described above for preliminary data supporting RNase titration effects. Collectively, the new fPAR-CLIP, PNK ³²P assay and updated RNP western blot data provide additional control experiments for validating PACCE.

Finally, the direct, unspecific interaction of protein with the fluorescent oligoribonucleotide as suggested by the reviewer is not likely given that labeling is enzyme mediated in the fPAR-CLIP method. The crosslinked RNA-protein complex is separated by a denaturing gel and non-covalent

interactions would be dissociated. Thus, non-specific interactions (i.e., non-enzyme mediated associations with proteins) would not be detected at the expected molecular weight of the RNP as the fluorescent molecule would not be attached.

Reviewer #4 (Remarks to the Author):

Heindel et al developed a chemoproteomic strategy to identify RNA-binding proteins and sites in living cells. The method is based on an electrophilic purine analogue probe that shows extensive labeling of cysteines in proteomes and the authors applied it in a competitive profiling manner when SILAC cells were labeled with 4SU and irradiated with UV to induce crosslinking between RNA and proteins. The rationale is that the 4SU crosslinking at the protein-RNA interface would interfere with the purine probe labeling and a cysteine site that is quantified with a high SILAC ratio would be a potential target site proximal to the RNA crosslinked site from which an RNA-binding protein can be discovered. They did thorough analyses on the list of proteins and cysteine sites from the profiling and biochemically characterized some targets that were previously unknown with RNA-binding activity. They finally demonstrated that a novel target EXOC4 has a RNA-binding activity and plays a role in RNP formation. Overall, I think the technical novelty of this method is quite clear, which is distinguishable with other profiling methods of RNA-binding proteins. Although the competitive chemoproteomic profiling of cysteines have been extensively reported before (which by the way I think the authors should reference), this is the first report to my knowledge of being applied for RNA-binding proteins. The profiling efforts have resulted in a couple of datasets that should be of broad interest to chemical biology and RNA biology community – namely, the overall proteome reactivity profile of the purine probe and the 4SU competitive profiling data of this probe for hunting RNA-binding proteins. However, my major concern is the data presentation, especially the graphic demonstration and the writing styles, which have already been raised by other reviewers. I don't think the manuscript has been improved in terms of these aspects and I still find it challenging to read through.

To make the paper easier to read, we have made substantial effort to cut down on the details and simplify for clarity. The result is a reduction in the total number of pages in the main text (from 17 to 15 pages) as well as the specific changes to figures as requested by the reviewer below. All changes can be found in the tracked changes document.

Key changes include updates to the results section on synthesis and characterization of CEPs, which has been trimmed to focus on the most important points to guide readers. Also, the results section on benchmarking CEPs for cysteine-reactive profiling in cells has been condensed to highlight proteomic activity, comparisons with existing DBIA probe, and to initially establish CEPs as a global chemoproteomic probe of purine-binding proteins. The latter point addresses the suggestion by Reviewer #4 to introduce CEPs as a broad cysteine-reactive probe with the ability to target the RBP subclass for a smoother transition to the PACCE section.

We also added references for competitive ABPP profiling of cysteines as suggested by the reviewer in the Discussion section:

“Importantly, the use of chemical probes for RBP profiling is needed to enable and streamline ligand discovery efforts through competitive activity-based protein profiling (ABPP) screening^{46,66}.”

And

“Akin to other chemoproteomic methods, PACCE is well positioned to screen for covalent ligands using competitive ABPP methods^{46,66} that can perturb RBPs with protein class- and binding site-selectivity across the human proteome⁴⁶.”

Here are some specific points:

1) Figure 1 and 2 are with too many texts and should be re-drawn with attractive graphics/cartoons/ for clarity.

We updated both figures. Components of figure 2 were integrated into the workflow. New versions are included in the revised manuscript and attached below:

Figure 1. Development of clickable electrophilic purines for chemical proteomic profiling

Figure 2. Schematic of Photo-Activatable-Competition and Chemoproteomic Enrichment (PACCE) methodology.

2) The rest of figures are also too crowded and each subfigure should have enough margins left from its surrounding subfigures and with proper font sizes (some of which are too large to be confusing, e.g., figure 5B and 5C)

We addressed both spacing and content issues on all main and supporting figures. Figures 3 and 5 are highlighted below as examples.

Figure 3 – We reformatted the Venn diagram for improved figure spacing and clarity. In addition, we color coded to remove complicated lines.

Figure 5 - We moved the western blots to the supporting information (Figure S18) to highlight the new fPAR-CLIP data. This greatly reduced the complexity while retaining key data for supporting RNA-binding capabilities of our RBPs of interest.

3) The writings made it hard to follow the overall logic flow of this manuscript. The result sections are overwhelmed with too many technical details such as list of cell line names, percentage values, as well as abbreviation names coined by authors themselves (e.g, SR>2, 1:1 control). The authors should make an effort to polish the writing to convey main messages in each section and refer to the key figures to help support such messages.

We moved a substantial fraction of the technical details from the main text to figures, figure legends and the methods section to improve readability of the manuscript. All the changes can be found as tracked changes in the revised manuscript.

4) Based on the current frame, I suspect that the authors first made the purine probe and hope it can achieve the profiling of cysteines at the RNA-protein interfaces alone. Then the competitive labeling was introduced to narrow down those cysteines that are perturbed by 4SU photocrosslinking. If so, my suggestion is to establish the purine probe just as a generic cysteine-reactive probe for chemoproteomic profiling in living cells and then established the whole “PACCE” workflow to focus on the RNA-binding proteins. As also suggested by other reviewers, it needs more evidences to support the purine probe as the cysteine-reactive probes of the RNA-interactomes as there are clearly other types proteins being pulled out by the purine probe. Only when it is used together with the photoactivatable 4SU competition, the probe maximized its impact in capturing key cysteines at the RNA-protein interfaces.

We thank the reviewer for this suggestion. Figure 1B has been updated to show profiling of the purine-binding proteome using CEP-mediated chemoproteomics. The result section “Electrophilic purines are cysteine-reactive probes of the RNA interactome” has been revised to “Benchmarking CEPs as

cysteine-reactive probes in living cells” and additional language has been added to highlight the broader activity of CEP probes in proteomes. Figure 1 title has been updated from “Development of activity-based probes for RNA-binding proteins” to “Development of clickable electrophilic purines for chemical proteomic profiling.”

We emphasize the enrichment for RNA binding proteins when we introduce the PACCE workflow in the next results section, “Quantifying protein-RNA interactions in cells by photoaffinity competition” as recommended by the reviewer.

5) Although there is no hard limit, 30 are abnormally large for the number of SI figures and it will clearly hurt the overall readability of this manuscript. The authors are suggested to consolidate some of these data.

We reduced the number of supplementary figures from 30 to 20 through the following changes to the previous figure versions:

- ***Figure S1 was removed and replaced by literature references.***
- ***Figure S3 and S4 were combined.***
- ***Figure S11 was removed and the data uploaded to Table S3.***
- ***Figure S12 and S14 were combined.***
- ***Figure S20 was removed because data already present in Table S5***
- ***Figure S21 and S22 were removed.***
- ***Figure S27 and S29 were combined.***

Reviewers' Comments:

Reviewer #3:

Remarks to the Author:

I thank the authors for the attempts to further validate PACCE and improve the manuscript. Regrettably, I believe that the additional data provided not only fail to address my concerns but, on the contrary, further substantiate the worries I have expressed. As elaborated in detail, the data compelled me to believe that PACCE is unsuitable for detecting RNA-binding regions due to inherent conceptual limitations. I tried to explain why and to describe alternative controls/approaches that could be implemented to prove me wrong. I am concerned that in its current state, the work may inadvertently contribute to an artificial inflation of the number and diversity of RNA-binding regions, potentially causing confusion within the field. Please, find my comments in blue directly in the attached rebuttal letter.

REVIEWER COMMENTS

Reviewer #2 (Remarks to the Author):

Reviewer's comments on the revised manuscript by Heindel et al. "Chemoproteomic capture of RNA binding activity in living cells"

First at all I would like to apologize for the very much delayed review of the revised version of the manuscript.

It is my opinion that the authors have significantly improved their manuscript. I am particularly happy that the comparison of their CEP probe with iodoacetamide is now more reliable. Furthermore, the authors have addressed my other points in a sufficient manner.

Therefore, I am happy to state that the revised version of the manuscript should be published in Nature Communications.

We thank the reviewer for such a positive review!

Reviewer #3 (Remarks to the Author):

The authors describe the development of a method termed Photo-Activatable-Competition and Chemoproteomic Enrichment (PACCE) for the global quantification of protein-RNA contacts in cells. As described in detail below, I am not convinced about the performance of PACCE. I divided my comments in three main conceptual areas.

1) Specificity of the CEP probes towards RBPs

The authors compare the activity of CEP and of general cysteine-reactive probes (e.g. DBIA). These analyses in my opinion are instrumental to assess the performance of the CEP probes. Unfortunately, I think that the overall conclusion of these comparisons, as well as of other presented data (see below), is that the proposed CEP probes are a good tool to target cysteines in cellular proteins (Figure S8), but that are not an excellent choice to study RBPs. While the number of RBPs identified using the CEP probes might be acceptable, I am not convinced about the specificity of the probes. Note that among CEP-target regions, there are 5232 hits mapping to proteins involved in "protein binding" and just 1102 mapping to proteins with "RNA binding" function (and among these two groups, the cluster "protein binding" displays in fact a much lower FDR) (Table S3, GO molecular function).

We should clarify that the 5232 hits to "protein binding" and 1102 hits to "RNA binding" proteins correspond to DBIA and not the CEP datasets. If we directly compare the "RNA binding" GO function (GO:0003723) between the probes, we find CEP shows further enrichment compared with DBIA (3-fold for CEP (found in Table S3, 'ALLMF' tab) versus 2-fold enrichment for DBIA (found in Table S3, 'DBIAMF' tab).

I thank the authors for the clarification. The overall concept is still that the term "RNA-binding" is similarly enriched with a general cysteine-reactive probe or CEP (see Figure 1C). This is now tacitly recognized by the authors in lanes 157-160: "Compared with the general cysteine-reactive probe iodoacetamide (IA) and specifically datasets using the desthiobiotin-tagged analog (DBIA40), protein function enrichments were largely comparable between probes with the exception of nucleic acid- and protein-binding that were specific for CEP and DBIA, respectively (Figure 1C)."

Also note that the percentage of identified hits in the group “RNA metabolism protein” displays only a very subtle increase from 12% to 14-15% in CEP-based studies relative to DBIA-based studies (Figure S9B).

We should re-emphasize that the strength of CEPs and PACCE is the ability to perform proteomic investigation of RBPs in live cells, which we (Figure 3A) and others in the RBP field (a comprehensive review provided in PMID: 29339797) have shown to be critical for capturing RBP activity. The ability of CEPs to capture RNA interaction sites on proteins is substantially improved in live cells compared with in vitro labeling using existing cysteine-reactive probes (3-fold increase for CEP vs DBIA, Figure 3A).

Unfortunately, Figure 3A lacks the results of the in-situ approach with a general cysteine-reactive probe. Is this feasible? Nevertheless, even when the in situ approach leads to a higher number of RNA-sensitive cysteines, I don't think these residues necessarily correspond to RNA interaction sites as claimed (see below).

Furthermore, DBIA-based datasets show a much stronger enrichment of proteins bearing bonafide RNA-binding domains (as helicase ATP-binding and helicase C-terminal) relative to CEP-derived datasets.

While this is true, we should point out that the CEP probe shows stronger enrichment of proteins containing the RRM, KH, and double-stranded (dsRNA) RNA-binding domains (Figure 1D). RRM, KH and dsRNA domains are well-validated and important RNA-binding domains with numerous reports in literature (PMID 15853797, 36689472, 8405383, 11743725, 15573138, 22918483).

This comparison also highlights an important point: despite a nearly 4-fold decrease in cysteine site coverage using CEP (6214 vs 24151 cysteine sites detected by CEP vs DBIA, respectively; Figure 1D), the CEP probe can enrich for RNA-binding domains to a comparable and in some cases enhanced degree compared with the state of art cysteine-reactive probe DBIA.

As stated in the current version of the manuscript, and despite of some nuances, I think that an overall fair conclusion is that both CEP and DBIA similarly enrich for RNA-binding domains (see lane 166-168: “Domain enrichment analyses also revealed comparable coverage of RBDs, as well as other functional protein domains, for both CEP and DBIA (Figure 1D and Table S3).”)

I think that the low specificity of CEP probes towards RBPs is not surprising considering that the employed probe(s) just modestly resemble the regions in cellular RNAs recognized by RBPs (that are several nucleotides long and that can result from RNA folding into specific secondary structures). In addition, the amino acid preferentially targeted by the CEP probes, cysteine, is not common among RNA-binding domains (it is in fact negatively represented) (PMID: 16790841).

Cysteines are underrepresented in canonical RNA binding domains but we believe this could be a strength for developing small molecule modulators of this protein class. A recent report demonstrated the utility of developing cysteine-reactive ligands to target the RBP NONO for altering the transcriptomes of cancer cells via ligand induced stabilization of RBP-RNA interactions (PMID 36864190). These findings provide important proof of concept that cysteines found in allosteric regions

of RBPs are ligandable (and potentially druggable) sites for pharmacological modulation. Our initial evaluation of cysteine site sensitivity to RNA crosslinking in relation to RNA-cysteine distance (Figure 4C) provides a resource for enabling future investigation of direct and allosteric modulators of RBP function.

I don't question the utility of targeting cysteines on RBPs to control their activity, I question the utility of PACCE to identify RNA-binding regions (see below).

2) Low efficiency of UV crosslinking of protein-RNA interactions in PACCE

Regarding the PACCE approach, besides the aforementioned considerations, there is a conceptual problem: UV crosslinking is inefficient for numerous RBPs. Available data show that as little as 5% of a given RBP can be crosslinked to RNA (PMID: 33332543; PMID: 31586594). That means that the reduction upon irradiation of the pool of cysteines "free" to react with the CEP probes is expected to be very subtle for these proteins, which would compromise the detection of RNA-interacting cysteines.

4SU is a well-established methodology used in the RBP and RNA community for the past ~60 years (PMID 5321899) and a search on PubMed will reveal hundreds of published reports using 4SU crosslinking for RBP profiling. PACCE could be improved if a more efficient crosslinking version of 4SU could be established in the field. However, 4SU remains the state of the art and development of a new crosslinkable RNA nucleoside is beyond the scope of the current manuscript.

I don't question the utility of 4SU, I question the conceptual design of PACCE to identify RNA-binding regions. I will try to explain myself in a different way: Regardless of the implementation or not of 4SU, just a small fraction of most cellular RBPs is crosslinked to RNA. This is shown in the literature, using both western blot and unbiased proteomic approaches (e.g. PMID: 33332543; PMID: 31586594), and it is also shown in this manuscript (Figure S18). In Figure S18 A-C, the alleged non-crosslinked fraction of the proteins PTBP1, EXOC4 and UIMC1 (which display the expected monomeric molecular weight), does not show any appreciable difference regardless of the application or not of UV or RNase. This suggests that just a minor pool of these proteins is crosslinked to RNA, which is further confirmed in Figure S18D-E, and in Figure 5B as the authors themselves state below in this rebuttal letter ("Please also note that we previously found that at the 4SU concentrations used for PAR-CLIP and fPAR-CLIP (100 μ M, 16 hrs pulse) we expect a substitution of 1 in 40 Us with 4SU (PMID: 20371350). This in turn means that only a small fraction of the RBP bound transcriptome will crosslink and thus, the 20 kDa shift due to adapter ligation will not be reflected in the Western Blots."). This low "crosslinkability" stands in stark contrast with the strong SILAC ratios obtained by PACCE (with PACCE hits displaying SILAC ratio > 2). Overall, I think the lack of consistency between PACCE ratios on one side, and the crosslinking efficiencies reported in the literature/observed by the authors on the other, indicates that PACCE does not report RNA-binding regions.

If the authors know that the crosslinking strategy they apply leads to much higher average efficiencies (e.g. due to the UV wavelength, use of 4SU, etc.), it would be useful to state it and show the supporting data. Otherwise, and for the exposed reason, I would assume that the utility of PACCE is rather narrow.

We considered several avenues for RNA crosslinking when developing PACCE and chose UV crosslinking at 312 nm using 4SU based on reports demonstrating these conditions can enhance specificity for protein-RNA crosslinks with minimal loss to overall proteome sensitivity (PMID

27768875 and 24141703). As the reviewer pointed out, formaldehyde has been used for investigating RBP-RNA interactions and while more efficient compared to UV, this chemical crosslinking technique can introduce additional artifacts through protein-protein (PMID 32561732, 17593931) and protein-DNA crosslinks (PMID 25101823, 10606672, 24848408), which would further confound our competition studies. There are ongoing discussions in literature regarding the advantages and disadvantages of using UV compared with formaldehyde for protein-RNA crosslinking, but a general consensus is that UV is more specific (PMID 30804549). Given that we are developing a new chemoproteomic methodology for RBP profiling, we prioritized specificity for our proof-of-concept studies. We should point out that PACCE is amenable to chemical crosslinking, and we can explore formaldehyde-mediated RBP-RNA interactions in future studies.

To further clarify our rationale for choosing 4SU-RNA crosslinking at 312 nm, we updated the results section in the revised manuscript as follows:

“We chose to incorporate 4SU for UV-mediated crosslinking of cellular RNA to protein at 312 nm because of (i) higher specificity (e.g., DNA-protein crosslinks and single-strand breaks are ~1000-fold less at 312 compared with 254 nm⁴²) without compromising proteomic sensitivity^{27,43}, (ii) less damage to cells^{1,27}, and (iii) a UV wavelength closer to the optimum extinction coefficient of 4SU⁴⁴ (Figure S4 and S9)”

We also updated the limitations/future directions section of the discussion as follows:

“The versatility of 312 nm to crosslink native and 4SU-RNA was an important consideration given the reported 4SU incorporation rates in cells (1-4% of uridines⁷³). Chemical crosslinking could increase efficiency for detecting RBP-RNA interactions but would require careful selection of controls to account for lower specificity due to protein-protein and protein-DNA crosslinks⁷⁴.”

It sounds good to me. The main point of my comment nevertheless, was to give the authors the chance to explain if the crosslinking efficiency in their protocol was much larger than the one typically seen in the literature. Fig S18 suggests that this is not the case, and that “crosslinkabilities” in this manuscript and the literature are alike, which questions the utility of PACCE to detect RNA-binding regions (see previous comment).

Note that the results presented in Figure S17 are of limited value, given the fact that RNase treatment is expected to cause massive changes in the crowding of the lysates as well as in the solubility of the proteins (PMID: 32945072). This could influence cysteine reactivity indirectly rather than by means of a competition between RNA and the probe, as suggested. The use of +/- UV irradiation in the absence of any RNase treatment might be a better option. Furthermore, these analyses should be done in cells, given that the authors detect 7 times more “RNA-sensitive” cysteines with in situ PACCE relative to the in vitro variant, and that that is the focus of the rest of the paper.

We do not believe indirect, proteome-wide effects are occurring in our RNase treatments. If RNase treatments were causing a massive, non-specific alteration in proteomes, the SILAC ratios of peptides (both probe-modified and non-modified peptides) would be heavily skewed (SR >>2 and/or SR <<2) in (+)RNase vs. (-)RNase conditions. We do not observe this type of effect. In fact, we determined the median SILAC ratio of all peptides detected in our RNase treated datasets, averaged this value across 3 biological replicates and found this value to be ~1.2.

Would such a skewed signal upon RNase treatment indeed be expected? It is hard to follow the exact protocol employed, but in methods under “RNase treatment”, it is stated that: “A desthiobiotin enrichment tag was then added using click-chemistry as described above, and the samples were processed using chloroform/methanol extractions as described above”. Then, in the “Preparation of proteomes for SILAC LC-MS/MS chemical proteomics” section it is stated that: “The insoluble pellet was resuspended in 6 M urea, 25 mM ammonium bicarbonate (ambic).” I thus understand that protein aggregates that could potentially result from the RNase-treatment could be re-dissolved in the urea-containing buffer and thus would not necessarily influence the overall protein intensity. Furthermore, even if the overall protein solubility is not affected, the RNase treatment could influence cysteine reactivity indirectly by means different than the alleged competition between RNA and the probe (e.g. by affecting molecular crowding, protein-protein and protein-ligand interactions). For these reasons, I had suggested to complement the presented data (+/- RNase treatment of lysates) with an analysis involving +/- UV irradiation of cells (without RNase treatment). Without this experiment, I would not conclude that RNA-sensitive cysteines correspond to RNA-binding regions.

The in situ RNase treatment is a potentially interesting experiment but will be confounded by indirect effects because of global re-localization of RBPs from the cytoplasm to the nucleus from RNase activation in cells (e.g., PMID 36318584). Thus, introduction of an active RNase to cells could influence cysteine reactivity indirectly because of the accessibility of RBPs to probe labeling and not strictly from competition between RNA and CEP probe.

Please note that I suggested a non-UV control (and not to introduce RNase to cells).

3) Lack of proper validation of PACCE

It would be essential to validate further and properly the capacity of PACCE to detect RNA-binding domain/residues, which ideally should involve multiple proteins. I find that the supposed validation currently presented in the paper (Figure 5) is NOT conclusive. The authors claim: “...we observed a shift in molecular weight upon UV irradiation of 4SU-labeled, EXOC4-expressing HEK293T cells that was not observed in the absence of UV irradiation and reduced with RNase treatment.” But I don’t think this is true; albeit weaker, the same boxed bands are still visible in non-irradiated samples, and it is not clear why.

We suspect crosslinking from ambient light is producing the higher molecular weight species observed for recombinant EXOC4-expressing HEK293T cells in the (-)UV condition. This behavior (i.e., cross-linked RNP signals in the absence of UV) has been reported for several known RBPs in literature reports (e.g., Figure 1F and G from PMID 30824702).

We have modified the results section as follows to clarify this point:

“Akin to the migration behavior of a known RBP (Polypyrimidine tract-binding protein 1 or PTBP1), we observed a shift in molecular weight upon UV irradiation of 4SU-labeled, EXOC4-expressing HEK293T cells that was muted in the absence of UV irradiation and reduced with RNase treatment (S18A and B). The higher molecular weight signals observed without UV irradiation are likely due to crosslinking from ambient light as previously reported⁵⁷.”

In addition, the RNase-response is extremely subtle and does not present the expected pattern (see PTBP1 in Fig 5B). Analogous suboptimal results are presented for the other chosen candidate, UIMC1.

A reduction but not complete ablation of RNA-binding signal with RNase treatment of RBPs is not uncommon in literature. For example, in the PTex report (PMID 30824702), the cross-linked RNP signals for known (PTBP1, FUS; Figure 1F and G) and candidate RBPs (ABCF2, CCT7; Figure 5H and I) are not completely ablated with RNase treatments. In some cases, only a subtle decrease was observed.

Note that the RNase treatment after PTex is expected to result in an increased and not a decreased signal at the monomeric molecular weight of the RBP, given that the digestion of the RNA should remove the smear of the RBPs. This is what is actually shown for FUS and GAPDH (Fig 1F), and HuR (Fig 1C) and what has been shown in other publications for multiple RBPs (e.g. PAB1, hnRNPC1/C2, GAPDH and FASTKD4 in PMID: 30035255). I acknowledge that the response of PTBP1 in PMID 30824702 is opposite than expected. Please note that the decreased FUS signal in Fig 1G (PMID 30824702) represents a different analysis (RNase treatment was performed before PTex). I wonder whether the responses of the candidate RBPs ABCF2, CCT7 in Figure 5H and I respond to the same reason (I failed to find in the paper whether the RNase treatment was performed before or after PTex). In any case, while some RBPs might lack a proper response to the RNase treatment for reasons that are not understood, I don't think such an outcome could be used to prove RNA-binding of alleged "novel" RBPs that lack any other convincing experimental validation. Thus, I consider that Figure S18 B-C do not prove RNA binding of EXOC4 and UIMC1. An orthogonal assay as RIC, eRIC, PTex, XRNAS, OOPS or 2C could be tried. Alternatively, other candidate RBPs (ideally more than 2) could be picked from the list of PACCE hits and studied. Some unconventional RBPs are just very hard to pursue experimentally.

We performed additional control experiments to verify protein-RNA interactions by the polynucleotide kinase (PNK) assay as previously reported (PMID: 30773316). Cellular lysates from UV (312 nm) irradiated, 4SU-RNA-treated HEK293T cells recombinantly expressing PTBP1, EXOC4 or UIMC1 were subjected to RNaseA at increasing concentrations. Afterwards, proteins were immunoprecipitated followed by radioactive labeling (³²P) of RNA 5' ends with T4 polynucleotide kinase and imaging of autoradiographic film. Western blotting was performed to confirm protein expression across samples.

As shown in the Figure below, we observed increased higher molecular weight radiolabeled bands upon UV irradiation of HEK293T cells overexpressing protein and this smeared signal corresponding to RNP complexes was reduced in a RNaseA-concentration dependent manner. Note that the protein loading for (-)RNase treatment sample of UIMC1 was higher than the other samples but RNaseA concentration dependence was still apparent when comparing 0.5 and 2 ng/μL RNaseA-treated samples.

Figure S18F. Control experiments to verify protein-RNA interactions by the polynucleotide kinase (PNK) assay.

We have included this new data as Figure S18F in the revised manuscript. The results section has also been updated as follows:

“We further verified protein-RNA interactions by the polynucleotide kinase (PNK) assay⁶³. Cellular lysates from UV (312 nm) irradiated, 4SU-RNA-treated HEK293T cells recombinantly expressing PTBP1, EXOC4 or UIMC1 were subjected to RNaseA at increasing concentrations. Afterwards, proteins were immunoprecipitated followed by radioactive labeling (³²P) of RNA 5' ends with T4 PNK. As shown in the Figure S18F, we observed increased higher molecular weight radiolabeled bands upon UV irradiation of HEK293T cells overexpressing RBP and this expected ‘smeared’ signal corresponding to RNP complexes was reduced in a RNaseA-concentration dependent manner.”

We recognize that the RNase response for UIMC1 is more subtle compared with PTBP1 and EXOC4 and thus performed additional biological replicates to quantify the reduced RNP signals for UIMC1. The new data show significant reductions in UIMC1 RNP signal upon RNase treatments and are included in revised Figure S18C and attached below:

Figure S18C. Updated western blot analysis showing RNP formation upon UV crosslinking in HEK293T cells expressing recombinant protein UIMC1 that was significantly reduced with RNase treatment.

I'm afraid that the inadequate quality of the provided PNK assay data prevents me from drawing any conclusions about the RNA-binding activity of EXOC4 and UIMC1. I thus consider that PACCE has not been properly validated, and thus unfortunately I am forced to question its utility to identify RBPs and RBDs.

In Figure 5C, note the occurrence of multiple bands in each lane, from which the authors select those that match the expected MW to claim RNA association. Given the lack of western blot analysis and proper controls (e.g. RNase titration), it is not possible to judge whether these bands correspond or not to the alleged proteins (or for example to contaminating RBPs), and it is not possible to judge neither whether the signal actually comes from labelled RNA (or, e.g., from a direct "unspecific" interaction of the protein with the fluorescent oligoribonucleotide; a type of interaction I have seen happening in gel).

We repeated the fPAR-CLIP experiment and included a mock-transfected (empty expression plasmid) control. As you can see in the revised Figure 5B, all background fluorescent bands (which may derive from "unspecific" interactions of proteins with fluorescent oligoribonucleotides) are present in the mock control, which is clearly differentiated from the specific fluorescent bands from crosslinking FLAG-PTBP1, FLAG-EXOC4 and FLAG-UIMC1 RNPs. Please note that for fPAR-CLIP the migration of crosslinked, adapter ligated RNPs is expected to run ~20 kDa above the expected migration of non-crosslinked RNPs (PMID: 33503264), as seen in the matching Western Blot in Figure 5B. Please also note that we previously found that at the 4SU concentrations used for PAR-CLIP and fPAR-CLIP (100 μ M, 16 hrs pulse) we expect a substitution of 1 in 40 Us with 4SU (PMID: 20371350). This in turn means that only a small fraction of the RBP bound transcriptome will crosslink and thus, the 20 kDa shift due to adapter ligation will not be reflected in the Western Blots.

Figure 5B. Updated fPAR-CLIP analyses with matching western blots.

We performed RNase treatments at high and low concentrations in the fPAR-CLIP studies as suggested by the reviewer (see below). Similar to observations with other RBPs (PMID: 33503264), the migration pattern of crosslinked and adapter-ligated RNPs did not change dramatically. As fPAR-CLIP uses high RNase concentrations for stringent footprinting, we typically only observe dramatic effects for RNase titrations when dealing with large complexes that contain multiple RBPs.

fPAR-CLIP studies performed with high (1.5 U/μL) and low (0.15 U/μL) RNase treatments.

We direct the reviewer to our PNK ³²P assay result described above for preliminary data supporting RNase titration effects. Collectively, the new fPAR-CLIP, PNK ³²P assay and updated RNP western blot data provide additional control experiments for validating PACCE.

Finally, the direct, unspecific interaction of protein with the fluorescent oligoribonucleotide as suggested by the reviewer is not likely given that labeling is enzyme mediated in the fPAR-CLIP method. The crosslinked RNA-protein complex is separated by a denaturing gel and non-covalent

interactions would be dissociated. Thus, non-specific interactions (i.e., non-enzyme mediated associations with proteins) would not be detected at the expected molecular weight of the RNP as the fluorescent molecule would not be attached.

Despite the sensible expectations based on the theory, surprisingly, it is not uncommon to observe non-covalent interactions in denaturing gels. For example, we have observed non-covalent interactions between RBPs and nucleotides in SDS-PAGE in our lab. As the provided data do not exclude that possibility and do not show any response to the RNase treatment, I suggest these results are interpreted with caution. The use of a non-irradiated control would be very useful. This control should show a strong reduction in the fPAR-CLIP signal. The opposite would indicate that the signal is luckily an artifact independent from RNA crosslinked to the proteins.

Reviewer #4 (Remarks to the Author):

Heindel et al developed a chemoproteomic strategy to identify RNA-binding proteins and sites in living cells. The method is based on an electrophilic purine analogue probe that shows extensive labeling of cysteines in proteomes and the authors applied it in a competitive profiling manner when SILAC cells were labeled with 4SU and irradiated with UV to induce crosslinking between RNA and proteins. The rationale is that the 4SU crosslinking at the protein-RNA interface would interfere with the purine probe labeling and a cysteine site that is quantified with a high SILAC ratio would be a potential target site proximal to the RNA crosslinked site from which an RNA-binding protein can be discovered. They did thorough analyses on the list of proteins and cysteine sites from the profiling and biochemically characterized some targets that were previously unknown with RNA-binding activity. They finally demonstrated that a novel target EXOC4 has a RNA-binding activity and plays a role in RNP formation. Overall, I think the technical novelty of this method is quite clear, which is distinguishable with other profiling methods of RNA-binding proteins. Although the competitive chemoproteomic profiling of cysteines have been extensively reported before (which by the way I think the authors should reference), this is the first report to my knowledge of being applied for RNA-binding proteins. The profiling efforts have resulted in a couple of datasets that should be of broad interest to chemical biology and RNA biology community – namely, the overall proteome reactivity profile of the purine probe and the 4SU competitive profiling data of this probe for hunting RNA-binding proteins. However, my major concern is the data presentation, especially the graphic demonstration and the writing styles, which have already been raised by other reviewers. I don't think the manuscript has been improved in terms of these aspects and I still find it challenging to read through.

To make the paper easier to read, we have made substantial effort to cut down on the details and simplify for clarity. The result is a reduction in the total number of pages in the main text (from 17 to 15 pages) as well as the specific changes to figures as requested by the reviewer below. All changes can be found in the tracked changes document.

Key changes include updates to the results section on synthesis and characterization of CEPs, which has been trimmed to focus on the most important points to guide readers. Also, the results section on benchmarking CEPs for cysteine-reactive profiling in cells has been condensed to highlight proteomic activity, comparisons with existing DBIA probe, and to initially establish CEPs as a global chemoproteomic probe of purine-binding proteins. The latter point addresses the suggestion by Reviewer #4 to introduce CEPs as a broad cysteine-reactive probe with the ability to target the RBP subclass for a smoother transition to the PACCE section.

We also added references for competitive ABPP profiling of cysteines as suggested by the reviewer in the Discussion section:

“Importantly, the use of chemical probes for RBP profiling is needed to enable and streamline ligand discovery efforts through competitive activity-based protein profiling (ABPP) screening^{46,66}.”

And

“Akin to other chemoproteomic methods, PACCE is well positioned to screen for covalent ligands using competitive ABPP methods^{46,66} that can perturb RBPs with protein class- and binding site-selectivity across the human proteome⁴⁶.”

Here are some specific points:

1) Figure 1 and 2 are with too many texts and should be re-drawn with attractive graphics/cartoons/ for clarity.

We updated both figures. Components of figure 2 were integrated into the workflow. New versions are included in the revised manuscript and attached below:

Figure 1. Development of clickable electrophilic purines for chemical proteomic profiling

Figure 2. Schematic of Photo-Activatable-Competition and Chemoproteomic Enrichment (PACCE) methodology.

2) The rest of figures are also too crowded and each subfigure should have enough margins left from its surrounding subfigures and with proper font sizes (some of which are too large to be confusing, e.g., figure 5B and 5C)

We addressed both spacing and content issues on all main and supporting figures. Figures 3 and 5 are highlighted below as examples.

Figure 3 – We reformatted the Venn diagram for improved figure spacing and clarity. In addition, we color coded to remove complicated lines.

Figure 5 - We moved the western blots to the supporting information (Figure S18) to highlight the new fPAR-CLIP data. This greatly reduced the complexity while retaining key data for supporting RNA-binding capabilities of our RBPs of interest.

3) The writings made it hard to follow the overall logic flow of this manuscript. The result sections are overwhelmed with too many technical details such as list of cell line names, percentage values, as well as abbreviation names coined by authors themselves (e.g, SR>2, 1:1 control). The authors should make an effort to polish the writing to convey main messages in each section and refer to the key figures to help support such messages.

We moved a substantial fraction of the technical details from the main text to figures, figure legends and the methods section to improve readability of the manuscript. All the changes can be found as tracked changes in the revised manuscript.

4) Based on the current frame, I suspect that the authors first made the purine probe and hope it can achieve the profiling of cysteines at the RNA-protein interfaces alone. Then the competitive labeling was introduced to narrow down those cysteines that are perturbed by 4SU photocrosslinking. If so, my suggestion is to establish the purine probe just as a generic cysteine-reactive probe for chemoproteomic profiling in living cells and then established the whole “PACCE” workflow to focus on the RNA-binding proteins. As also suggested by other reviewers, it needs more evidences to support the purine probe as the cysteine-reactive probes of the RNA-interactomes as there are clearly other types proteins being pulled out by the purine probe. Only when it is used together with the photoactivatable 4SU competition, the probe maximized its impact in capturing key cysteines at the RNA-protein interfaces.

We thank the reviewer for this suggestion. Figure 1B has been updated to show profiling of the purine-binding proteome using CEP-mediated chemoproteomics. The result section “Electrophilic purines are cysteine-reactive probes of the RNA interactome” has been revised to “Benchmarking CEPs as

cysteine-reactive probes in living cells” and additional language has been added to highlight the broader activity of CEP probes in proteomes. Figure 1 title has been updated from “Development of activity-based probes for RNA-binding proteins” to “Development of clickable electrophilic purines for chemical proteomic profiling.”

We emphasize the enrichment for RNA binding proteins when we introduce the PACCE workflow in the next results section, “Quantifying protein-RNA interactions in cells by photoaffinity competition” as recommended by the reviewer.

5) Although there is no hard limit, 30 are abnormally large for the number of SI figures and it will clearly hurt the overall readability of this manuscript. The authors are suggested to consolidate some of these data.

We reduced the number of supplementary figures from 30 to 20 through the following changes to the previous figure versions:

- ***Figure S1 was removed and replaced by literature references.***
- ***Figure S3 and S4 were combined.***
- ***Figure S11 was removed and the data uploaded to Table S3.***
- ***Figure S12 and S14 were combined.***
- ***Figure S20 was removed because data already present in Table S5***
- ***Figure S21 and S22 were removed.***
- ***Figure S27 and S29 were combined.***

Reviewer #4:

Remarks to the Author:

I think the authors did a good job in addressing my concerns and suggestions. I support its publication in the current form.

REVIEWERS' COMMENTS

Reviewer #3 (Remarks to the Author):

I thank the authors for the attempts to further validate PACCE and improve the manuscript. Regrettably, I believe that the additional data provided not only fail to address my concerns but, on the contrary, further substantiate the worries I have expressed. As elaborated in detail, the data compelled me to believe that PACCE is unsuitable for detecting RNA-binding regions due to inherent conceptual limitations. I tried to explain why and to describe alternative controls/approaches that could be implemented to prove me wrong. I am concerned that in its current state, the work may inadvertently contribute to an artificial inflation of the number and diversity of RNA-binding regions, potentially causing confusion within the field. Please, find my comments in blue directly in the attached rebuttal letter.

While we thank the reviewer for the additional critiques, we respectfully disagree that PACCE is not suitable for detecting RNA-binding regions and will cause confusion within the field. We remain enthusiastic that PACCE will complement existing proteomic methods to advance basic understanding of RNA-binding proteins (RBPs) and enable global screening for small molecule ligands of RBPs for pharmacology and drug discovery.

Reviewer #4 (Remarks to the Author):

I think the authors did a good job in addressing my concerns and suggestions. I support its publication in the current form.

We thank the reviewer for supporting publication of our work!